# Tunable templating of photonic microparticles via liquid crystal order-guided adsorption of amphiphilic polymers in emulsions

Xu Ma [1], Yucen Han [2], Yan-Song Zhang [1], Yong Geng [1], Apala Majumdar[2] & Jan P. F. Lagerwall [1] ✉

Multiple emulsions are usually stabilized by amphiphilic molecules that combine the chemical characteristics of the different phases in contact. When one phase is a liquid crystal (LC), the choice of stabilizer also determines its configuration, but conventional wisdom assumes that the orientational order of the LC has no impact on the stabilizer. Here we show that, for the case of amphiphilic polymer stabilizers, this impact can be considerable. The mode of interaction between stabilizer and LC changes if the latter is heated close to its isotropic state, initiating a feedback loop that reverberates on the LC in form of a complete structural rearrangement. We utilize this phenomenon to dynamically tune the configuration of cholesteric LC shells from one with radial helix and spherically symmetric Bragg diffraction to a focal conic domain configuration with highly complex optics. Moreover, we template photonic microparticles from the LC shells by photopolymerizing them into solids, retaining any selected LC-derived structure. Our study places LC emulsions in a new light, calling for a reevaluation of the behavior of stabilizer molecules in contact with long-range ordered phases, while also enabling highly interesting photonic elements with application opportunities across vast fields.

Introduced only 16 years ago[1], liquid crystal shells—a double emulsion where a middle phase of hydrophobic liquid crystal (LC) forms a thin self-closing spherical layer that surrounds, and is surrounded by, isotropic aqueous solutions—have emerged into a prolific platform for conducting stimulating fundamental physics research[2,3] as well as making innovative and broadly applicable photonic materials[4–6]. The latter activities focus strongly on cholesteric LC (CLC), also called chiral nematic, shells and droplets that exhibit omnidirectional wavelength- and polarization-selective retroreflection, giving rise to intriguing colorful patterns[7–13] that can be dynamically modulated[14–17], enabling omnidirectional lasing[18,19], as well as providing a handle to manipulate the particles via their interaction with light[19–23]. These

highly useful phenomena arise because CLCs self-organize with a helical modulation of period (pitch) $p_0$ of the director $\mathbf{n}$ (the average orientation of the LC-forming molecules, called mesogens) along an axis $\mathbf{m}$ that is perpendicular to $\mathbf{n}$, see Fig. 1a.

Because LCs are birefringent with $\mathbf{n}$ equal to the optic axis, the helical modulation leads to a periodic variation of the effective refractive index along $\mathbf{m}$ that causes Bragg diffraction of light with wavelength $\lambda = \bar{n}\cos\theta$, where $\bar{n}$ is the average refractive index in the CLC and $\theta$ is the angle of incidence with respect to $\mathbf{m}$. The reflected light is circularly polarized with the same handedness as the helix. The fact that $p_0$ can easily be tuned by varying the composition of the CLC mixture means that we can choose the retroreflection ($\theta = 0$)

[1]Experimental Soft Matter Physics group, Department of Physics & Materials Science, University of Luxembourg, 1511 Luxembourg, Luxembourg. [2]Department of Mathematics and Statistics, University of Strathclyde, Glasgow, United Kingdom. ✉e-mail: Jan.Lagerwall@lcsoftmatter.com

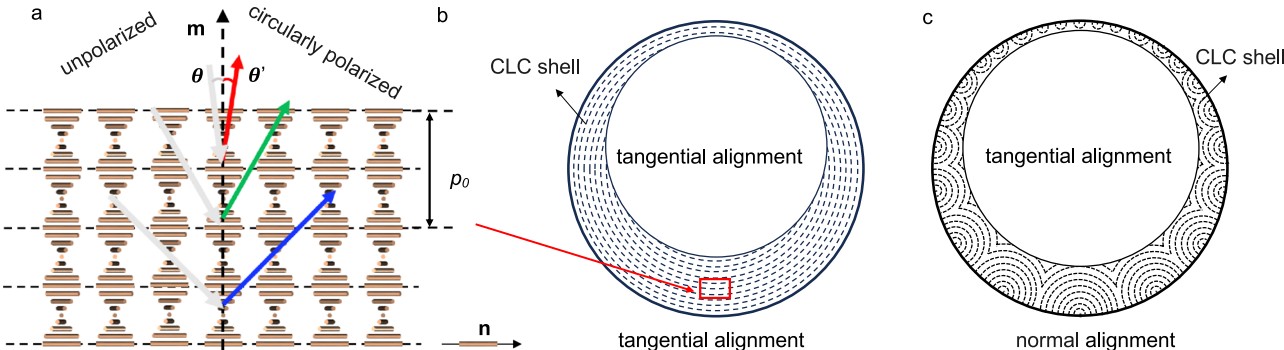

**Fig. 1 | Cholesteric selective reflection and ordering within shells with different boundary conditions. a** Schematic illustration of the local helicoidal order modulation in cholesteric liquid crystals (CLCs), defining the director **n**, the helix axis **m** and the helix pitch $p_0$. The wavelength- and polarization-selective Bragg diffraction is illustrated for three different angles of incidence $\theta$. Cross sections of cholesteric shells are drawn schematically for tangential (**b**) and hybrid (**c**) boundary conditions, in the latter case ignoring the topographical surface modulation found in the experiments. The distance between consecutive dashed lines is $p/2$ and **m** is everywhere normal to these lines. The density of the inner isotropic aqueous solution droplet is assumed lower than that of the LC, driving the droplet upwards in gravity and giving the shell a thin top and thick bottom.

wavelength at will. Finally, by using reactive mesogens (RMs), the shells or droplets can be turned into solid and durable Cholesteric Spherical Reflector (CSR) particles by photopolymerization once the CLC order has reached its equilibrium state[24–28]. The unique optical properties of CSRs have stimulated proposals for applications across diverse fields, from replacing traditional absorption-based pigments by structural color[25,26,29], even generating non-spectral colors like white[30], to invisibly encoding information onto surfaces[24] for linking physical objects securely to their digital twins[31] and thwarting counterfeiting[8,9,32], or providing human-invisible navigation support for robots and augmented reality devices[4].

The spherical symmetry giving rise to omnidirectional selective retroreflection is achieved by preparing the CLC shells such that **m** is oriented radially. This is done by imposing boundary conditions that force **n** to be in the plane of the CLC–water interfaces (tangential—also called planar—alignment), thereby promoting radial **m** (since **m** ⊥ **n**). The traditional way of achieving this is to stabilize the shells using a water-soluble polymer such as polyvinylalcohol (PVA), which should not enter the LC and thus allow water in contact with the LC to promote tangential **n**[33]. If normal alignment (**n** perpendicular to the interface; for flat samples often called homeotropic alignment) is desired, ionic surfactants are often used. However, boundary conditions at an interface between an LC and an isotropic liquid solution are complex, with surprising variations reported. In fact, ionic surfactants can give both tangential and hybrid (tangential on one side and normal on the other) alignment if the surfactant concentration is kept low[34], and polymeric stabilizers were recently found to change their aligning influence from tangential to normal as the shells are heated close to the LC–isotropic transition[35,36]. Very recently we also demonstrated that the chemical nature of the mesogens has great impact on both shell stability and alignment[37].

In this work we make use of the temperature-dependent boundary conditions enabled by amphiphilic polymeric stabilizers to dynamically tune the configuration of CLC shells with $p_0$ in the range for visible Bragg diffraction, from radial **m** with omnidirectional Bragg diffraction[4,8] at low temperature to a modulated polygonal texture with (imperfect) focal conic domain (FCD) configuration[2,38,39] at elevated temperature, or vice versa. The highly regular packing of FCDs has been well studied in smectic-A (SmA) LCs[40–42], also on curved substrates[43–46], but SmA phases do not exhibit Bragg diffraction of visible light. Cholesteric FCDs are less studied, and CLC shells with FCDs were investigated only with $p_0$ much too long to give visible reflections[39]. Moreover, the ease of our method in continuously tuning the configuration offers an unprecedented level of structural control. We propose a new model for explaining the tunability, for the first time

considering the entropic impact of the LC orientational order on the conformational freedom of water-dissolved amphiphilic polymers used to stabilize LC shells. This leads to a feedback loop where the stabilizer influences the LC configuration, but the LC order also influences the stabilizer molecules' behavior, reverberating back to the LC in terms of boundary conditions that change with temperature.

Using short-pitch CLCs we obtain a self-assembled structure that is a topographically inverted analogue to the intricate polygonal texture on the cuticle of certain beetles, responsible for their spectacular reflective colors[47,48]. The experimental results are complemented by a mathematical modeling and numerical simulation of cholesteric ordering in shells with different combinations of boundary conditions, reproducing the experimental findings. By using reactive mesogens we transform the LC shells into solid particles by photopolymerization at any temperature of our choice, preserving either the radial helix or the FCD configuration, exhibiting very different photonic functionality.

## Results

### Texture development on heating toward the clearing transition
To demonstrate the dynamic alignment tuning we first produce shells (Supplementary Fig. 1) of a polymerizable CLC base mixture using a standard aqueous solution of PVA (86–87% hydrolyzed; see Methods for further details) for the surrounding isotropic phases. After annealing until the shells are nearly defect-free, we heat them from room temperature to $T_{N^*I} \approx 72.4\,°C$ (we use $T_{N^*I}$ to indicate the onset on heating of the transition, which extends over a range of a few degrees since we are using multicomponent CLC mixtures, see Supplementary Fig. 2 and Supplementary Table 1). The process is monitored with polarizing optical microscope (POM) in transmission mode, as shown in Supplementary Movie 1, representative snapshots focusing on a single shell shown in Fig. 2. The aqueous PVA solutions impose tangential alignment of the CLC shell at room temperature, as recognized by a texture (Fig. 2a) that is characteristic of tangential short-pitch CLC shells viewed in transmission[19,49]. As the temperature is raised, the shell texture remains qualitatively unchanged up to $T \approx 70\,°C$, see Fig. 2b, the only significant change being a reduction in the number of interference rings which can be understood as a result of the decreasing birefringence upon heating[49,50].

However, at an alignment transition onset temperature $T_t = 70.8\,°C$ a qualitative texture change is seen, with radial striations first appearing along the circumference, see panel (c). As the temperature is further raised to the vicinity of $T_{N^*I}$, multiple polygons appear, as recognized in Fig. 2d–f. In (d), a transient texture reminiscent of soliton-like structures reported for CLC shells with much longer pitch[51] can be seen temporarily. This suggests[38] that the initially

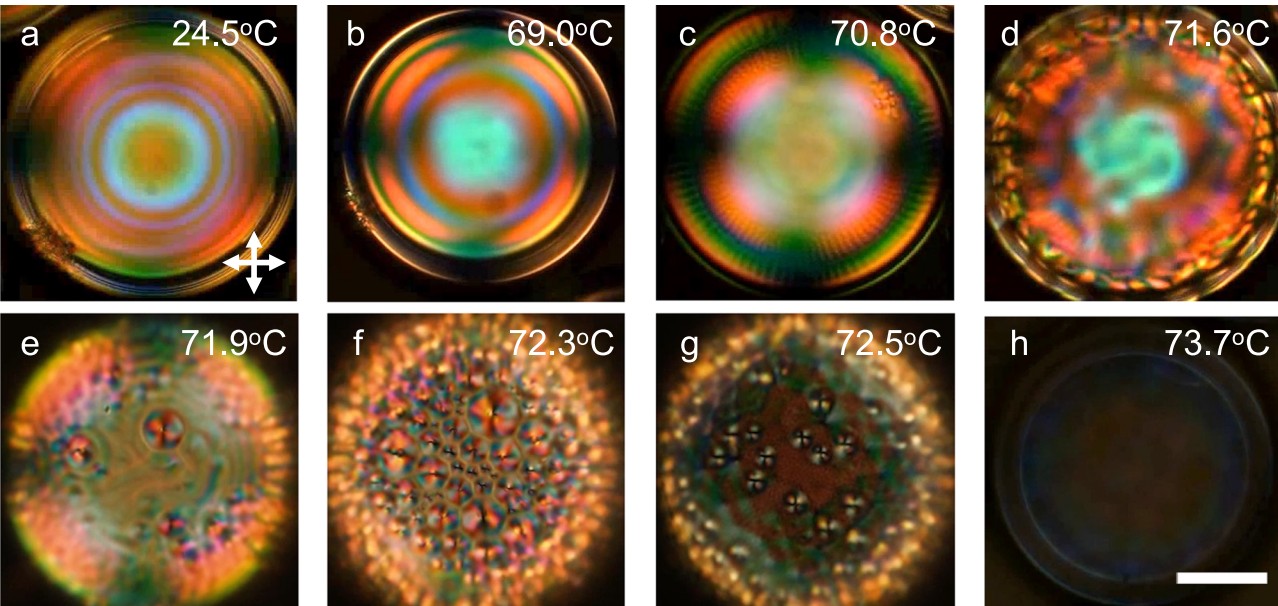

**Fig. 2 | Alignment transition upon heating a cholesteric liquid crystal (CLC) shell stabilized by polyvinylalcohol (PVA).** Transmission POM images (from Supplementary Movie 1) of a shell of the CLC base mixture stabilized by 86–87% hydrolyzed PVA heated from room temperature to $T = 73.7$ °C, where it is fully isotropic. The heating rate changes across three ranges: 5 °C/min for $T = 25$–65 °C, 1 °C/min for $T = 65$–69 °C, and 0.3 °C/min for $T = 69$–74 °C. The shell is between crossed polarizers (orientations indicated in **a**) and the focus is at the equator of the shell in **a**–**d**, then changing to the bottom surface in **e**–**h**. The brightness of the photo in **h** has been digitally enhanced to facilitate the detection of the shell boundary, otherwise difficult since the shell is here isotropic. Scale bar: 50 μm.

tangential boundary conditions change to normal at at least one interface near the clearing transition. When the temperature reaches the window of clearing, about $T = 72.4$–73.7 °C, the domains gradually disappear and the texture gets increasingly dark (g), until the shell can hardly be recognized between crossed polarizers because it is fully isotropic (Fig. 2h). Unfortunately, the shells break almost immediately when they are again cooled down to the cholesteric state (this is further discussed toward the end of the article). For the purpose of facilitating comparisons with the experiments to follow, we calculate a reduced temperature for the onset of the alignment transition, stating all measured temperature values in Kelvin, as $T_r = \frac{T_t - T_{N^*I}}{T_{N^*I}} \approx \frac{343.8 - 345.3}{345.3} \approx -0.0043$.

## Tuning the mixture for optimized phase sequence and viscosity
The physical behavior of LC mixtures can be greatly modified by changing the composition or adding or removing certain components, the impact depending on the specific interactions between the mixed compounds[52]. Adding a low molar mass flexible molecule like 1,6-hexanediol diacrylate (HDDA) disturbs ordering, hence it lowers both melting and clearing points, and it reduces the effective shear viscosity experienced during flow in a microfluidic device. Both effects are useful to us, because the effective shear viscosity of the basic CLC mixture has a rather high value of 1085 mPa s at room temperature, and its melting range is near room temperature (Supplementary Fig. 3a). This requires us to heat all liquids and the glass capillary microfluidic device to successfully produce shells with this mixture, whereas HDDA-doped mixtures should allow processing at room temperature. The dual acrylate termination of HDDA ensures that it will become part of the solid CSR achieved by photopolymerization[53].

The expected reduction in $T_{N^*I}$ is confirmed by Differential Scanning Calorimetry (DSC) (Supplementary Fig. 3b), monotonically decreasing with increasing HDDA content from the original $T_{N^*I} \approx 72.4$ °C of the CLC base mixture to $T_{N^*I} \approx 29.4$ °C with 11% HDDA, yet with negligible impact on the CLC reflection color. With increasing HDDA concentration, the peak in the heat flow curve broadens, reflecting an expanding temperature range of the phase transition. The

reduction in viscosity is also confirmed, see Supplementary Fig. 4. We find an apparently exponential decrease as a function of HDDA mass fraction.

As a representative example of HDDA-doped CLC shells, we show textures of a shell with 6% HDDA surrounded by our standard PVA solutions in Supplementary Movie 2, snapshots of which are shown in Supplementary Fig. 6. Upon heating toward $T_{N^*I}$, the textural behavior is qualitatively identical to that seen in Fig. 2, but the significant difference is that all changes take place at much lower temperature. The alignment transition starts at $T_t = 45.8$ °C and clearing starts at $T_{N^*I} \approx 48.5$ °C, yielding a reduced temperature for the onset of the alignment transition $T_r = \frac{T_t - T_{N^*I}}{T_{N^*I}} \approx \frac{318.8 - 321.5}{321.5} \approx -0.0084$. This is about twice the magnitude of when no HDDA was present in the CLC mixture.

The reduction in temperature of the textural change reflects the reduction in clearing temperature range shown in Supplementary Fig. 3b. Because the clearing is near room temperature at 10–11% HDDA, annealed shells made with these mixtures exhibit an FCD texture already at room temperature, see Supplementary Fig. 7. To probe the impact of shell radius and, in particular, thickness, we prepare three series of shells with a 6% HDDA CLC mixture and the same aqueous PVA solution for the surrounding phases using a different emulsification equipment that yields a radius about 1/2 of that in Fig. 2, with three different thicknesses. We study the textural change as a function of temperature, as shown in Supplementary Fig. 8, finding a slight increase of $T_t$ with increasing thickness. The change in radius in this experiment does not show significant impact.

## Polymerizing the shells at different temperatures to lock in different configurations
During the temperature-driven realignment process, UV light can be used to initiate and drive the polymerization reaction of LC monomers and HDDA at a selected temperature in order to preserve the CLC shell structure in any state we desire, thereby making a variety of CSR types. To make CSRs with FCD configuration, we first anneal shells made with the 6% HDDA mixture at room temperature until they exhibit a nearly uniform tangential texture (Fig. 3a), and then heat to 47.4 °C where the

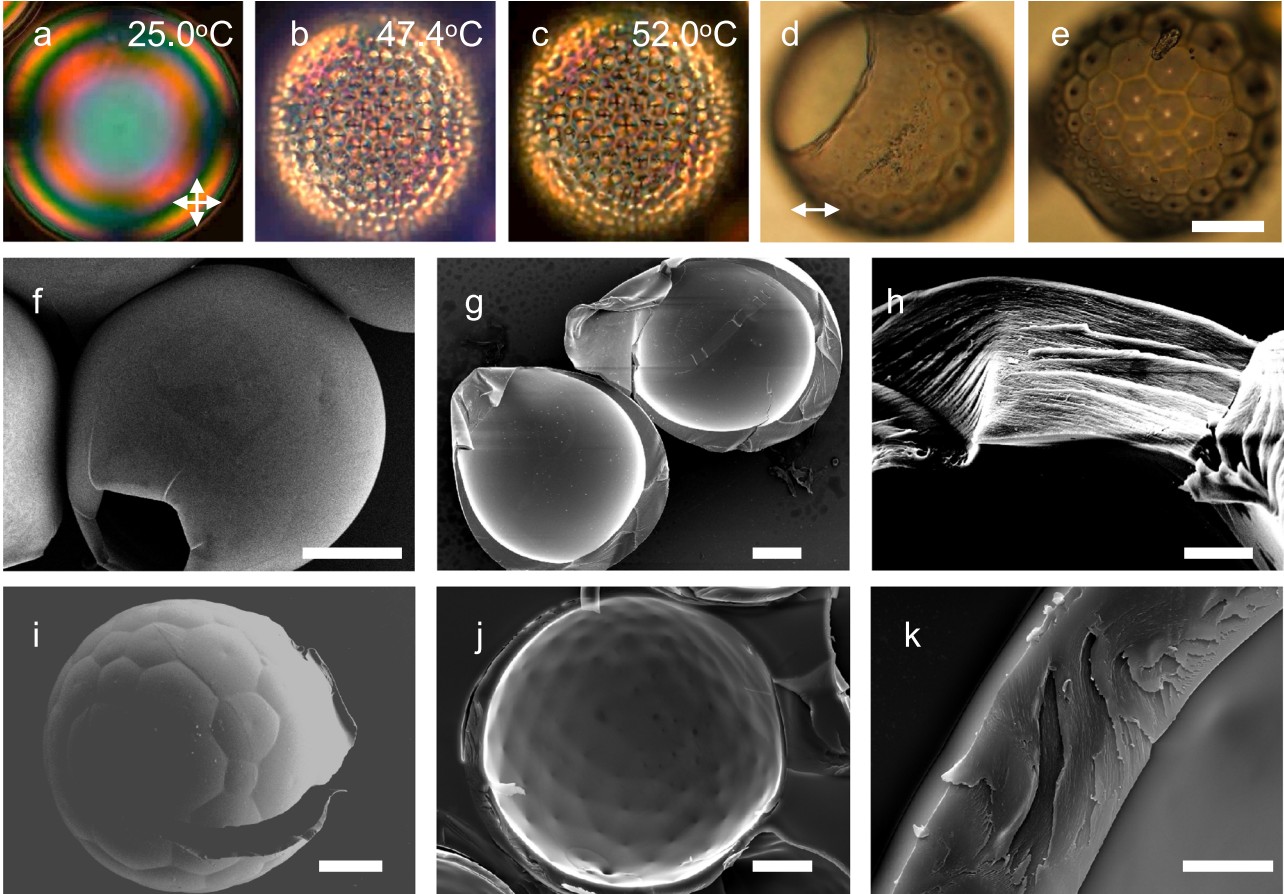

**Fig. 3 | Cholesteric liquid crystal (CLC) shells polymerized at temperatures of different alignment. a–c** Transmission POM images (polariser orientations indicated in **a**) of a CLC shell produced with a 6 wt.% HDDA mixture as it is heated from room temperature to 52 °C. UV light is applied to the shell at 47. 4 °C for 180 s, initiating polymerization. The heating rate was separated in four ranges: 5 °C/min for 25 − 45 °C, 0. 3 °C/min for 45–47 °C, 0. 1 °C/min for 47–48 °C, and 1 °C/min for 48–52 °C. The focus is at the equator of the shell in **a** and at the bottom surface in **b** and **c**. The polymerized FCD CSRs in **d**, **e** are imaged after coating with gold in transmission without analyzer (polarizer orientation in **d**). Scale bar: 50 μm. **f–k** SEM images of two gold-coated CSRs, with 9% (**f–h**) and 10% (**i–k**) HDDA, respectively. Both were polymerized at room temperature, yielding tangential boundary conditions for the former and hybrid for the latter. Images (**f/i**), (**g/j**), and (**h/k**), respectively, are obtained on the outer, inner and cross section surfaces of each CSR. Scale bar on images (**f**, **g**) and (**i**, **j**) is 50 μm, and on images (**h/k**) 10 μm.

shells exhibit an FCD texture. We now apply UV light for 180 s to polymerize the shells into CSRs, see Fig. 3b and Supplementary Movie 3. If the tangential configuration is desired, the same UV irradiation procedure is instead carried out at room temperature directly after annealing. Following polymerization, even with temperature increasing past the clearing transition of the CLC precursor mixture, the CSRs retain the exact same texture as when polymerization occurred, see Fig. 3c for the case of a CSR polymerized from a shell in the FCD state.

The CSRs are washed with water and acetone following the procedure introduced by Geng et al.[24]. Because the polymerization induces shrinkage of the shell, it bulges out slightly at the thinnest point as it must still encapsulate the same volume of incompressible aqueous solution inside. The stress induced upon acetone swelling ruptures the bulge, leaving a single hole in each CSR that allows complete removal of PVA from the inside, leaving a clean surface of the polymerized CLC material. Two such CSRs are shown after gold coating in bright field optical transmission microscopy from the side in Fig. 3d–e, clearly revealing how the size of the FCDs smoothly decreases from the thick to the thin side. One of the CSRs is sufficiently thin near the rim of the opening that the FCD formation appears to have been suppressed in that region, as indicated by the absence of visible FCD patterns. This continuous domain size variation may render CSRs polymerized from asymmetric shells with FCD texture attractive for photonics

applications, e.g., as resonators for microlasers[13], microactuator and sensors[16], or microprobes for optical trapping and manipulation[20,22].

The solid state of the polymerized CSRs allows us to image them by Scanning Electron Microscopy (SEM) after coating with a thin layer of gold. The results are shown in Fig. 3f–k. These images provide a deeper understanding of the director field configuration in shells with different texture, as well as of the shell surface topography. In a shell polymerized during tangential alignment both the inner and outer surfaces are smooth, as shown in Fig. 3f, g. In the cross section in Fig. 3h, visible pitch lines parallel to the inner and outer surfaces are seen, arising where the director aligns perpendicular to the fracture surface, leading to protrusions and holes since covalent bonds are broken during the fracture[54]. The orientation of these lines indicate a radial helix axis, normal to the CSR boundaries, thus with tangential boundary conditions for **n**. If the shell is instead polymerized under conditions of FCD formation, we recognize a regular array of protruding polygons with a recessed dot in the center on the outer surface, see Fig. 3i. On the inner surface the polygons are replaced by a diamond lattice with a recessed dot at each vertex, as shown in Fig. 3j. The outer and inner surfaces appear to have a topography modulated according to the hybrid director field sketched in Fig. 1c. The zoomed-in image of the cross section of this CSR (Fig. 3k) reveals undulating pitch lines, as expected for a director field breaking up into FCDs.

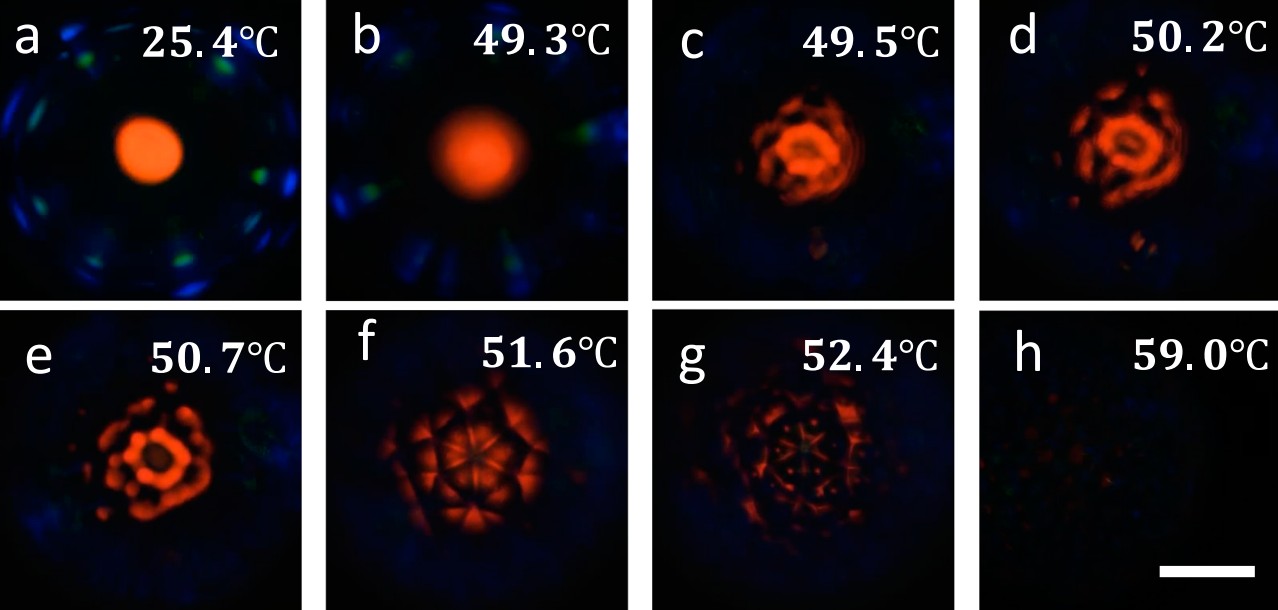

**Fig. 4 | Change in reflection behavior as a cholesteric liquid crystal (CLC) shell is heated through the alignment transition.** Reflection POM images of a CLC shell produced with 6 wt% HDDA mixture suspended in and surrounding isotropic PVA solutions (1.5 wt.%, 86–87% hydrolyzed) in glycerol (70 wt%) and water, as it is heated from room temperature until the shell turns isotropic. Temperature values are those reported by the hot stage, which are higher than at the sample because the experiment required the hot stage to be operated with open lid (Supplementary Fig. 5). The heating rate was separated in three ranges: 5 °C/min for 25.4–47.0 °C, 1 °C/min for 47.0–49.0 °C, and 0.3 °C/min for 49.0–61.0 °C. The focus is at the shell equator in **a**, **b** and at the top surface in **c**–**h**. Scale bar: 50 μm.

## Reflection behavior during alignment change and from FCD CSRs

The unique reflective properties of cholesteric LC shells showcase their tremendous potential for various applications. The properties are qualitatively very different for thin- and thick-topped shells[55]. As our CLC mixtures are denser than the aqueous solutions used so far, we have a thin top, yielding comparatively weak external reflection and allowing light into the interior of the shell, where it experiences a sequence of internal selective reflections giving rise to a characteristic ring-shaped pattern[55]. To instead emphasize the external reflections we need a thick-topped shell, hence we change both the inner and outer phases to a 1.5 wt.% solution of PVA (same type as before) in a mixture of 30 wt.% water and 70 wt.% glycerol to give it greater density than that of the CLC. The internal droplet then sinks down in gravity, yielding a thick-topped shell which gives strong reflection when we observe the shells from the top in the POM.

In Fig. 4 (snapshots from Supplementary Movie 4), we show the reflection behavior between crossed polarizers of such a thick-topped shell of a CLC mixture with 6 wt.% HDDA as we heat it from room temperature to 59 °C according to the hot stage, at which the shell is fully isotropic. This temperature reading for $T_{N^*I}$ is higher than in the corresponding experiment when the shell is observed in transmission, but this is an artifact arising from the way the experiment is conducted. To obtain a maximally rich reflection behavior, the shells are in a droplet of the PVA solution on a glass slide with a top interface to air. This leads to water evaporation and fogging of the hot stage cover window during regular operation, hence the stage must be left open. As shown in Supplementary Fig. 5 there is a linear correlation between the nominal setting with open hot stage (indicated in Fig. 4) and the real sample temperature; a recalibration confirms that the actual transition temperatures are practically identical to those seen for closed hot stage operation, as discussed in the Supplementary Information.

The top interface to air is important for the reflection behavior at room temperature, because the lower refractive index of the bounding medium means that we get three types of reflections from the shell[8]: a retroreflection spot with red color in the middle, a discontinuous ring of blue spots arising from cross communication directly between adjacent shells, and a discontinuous ring with slightly smaller radius of green spots arising from cross-communication mediated by a Total Internal Reflection event at the water–air interface. As the shell is heated, the reflection pattern remains quite constant up to 47 °C (hot stage setting), but upon further heating (b) we clearly see that the central spot as well as the cross communication spots get blurred. The alignment transition has started, and at 49.5 °C (c) we recognize the break-up into FCDs also in the retroreflection spot. As can be seen in Supplementary Movie 4, the texture now becomes extremely sensitive to the focus, with even fine adjustments significantly changing the pattern. The cross communication remains active but the resulting pattern is almost impossible to distinguish if the focal plane is at the retroreflection from the shell top. Upon further heating the top reflection gradually expands into a flower-like arrangement which in one focal plane (d–e) is reminiscent of the appearance of the FCDs in the cuticle of the beetle Chrysina Gloriosa in dark field microscopy[47], but in a slightly different focal plane (f) shifts to a set of six radial reflection lines. Upon further heating, the reflection texture loses both regularity and intensity at 52.4 °C, indicating that the transition to isotropic phase has started. At 53 °C the red reflection can hardly be recognized and at 59 °C (h) the isotropic state of the shell is confirmed by an entirely black image.

We also study the reflection behavior of shells polymerized into solid CSRs, see Fig. 5. The CSRs are made from the mixture used in Fig. 4 as well as from one with slightly lower chiral dopant concentration, yielding infrared (IR) retroreflection prior to polymerization. The shells are photopolymerized below the clearing point but after the transition to FCDs, and the CSRs are surrounded by air or in the UV-curable isotropic and transparent binder NOA 160 during imaging. Because the washing leads to some shrinkage of the structure, and thus reduction of $p_0$, the retroreflection from CSRs derived from shells like that in Fig. 4 is now green–yellow rather than red, while that of CSRs made from the originally IR-retroreflecting shells is red. We find a quite regular reflection pattern also in the polymerized FCD

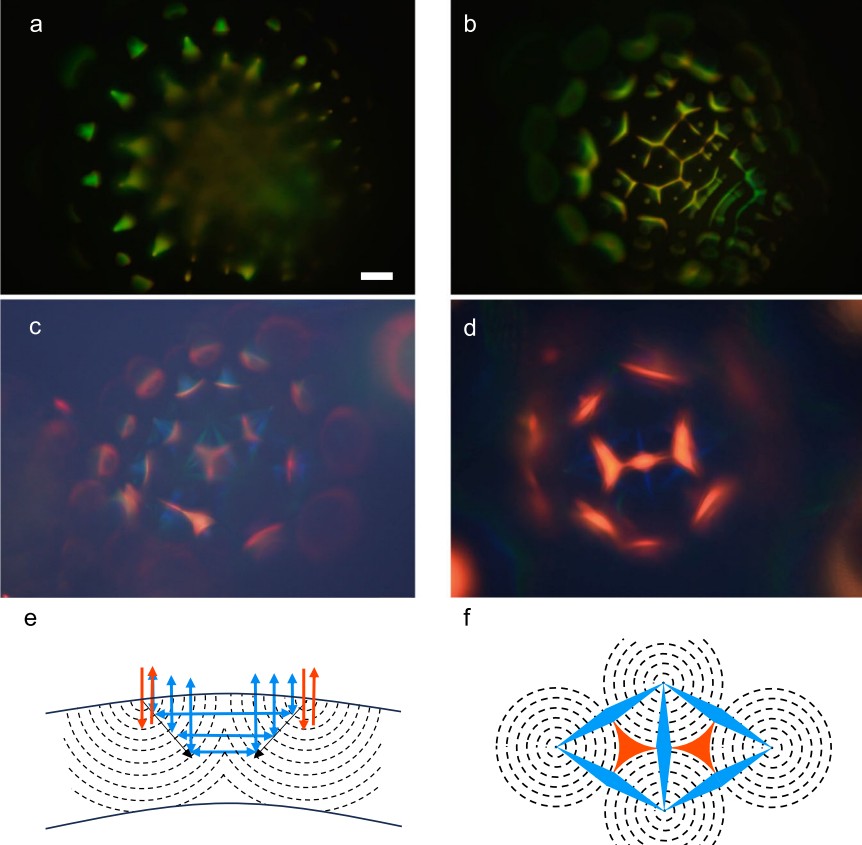

**Fig. 5 | Shell-internal photonic cross communication seen in polymerized cholesteric liquid crystal (CLC) shells with focal conic domain (FCD) configuration.** Reflection POM images of polymerized and washed CSRs, made from (**a**, **b**) the same CLC as in Fig. 4 and (**c**, **d**) from one with slightly longer pitch. The CSR in **a**, **b** is surrounded by air, with focus at the equator in **a** and at the top surface in **b**. The shells in **c**, **d** are embedded in NOA 160. Schematic drawings explaining the FCD-FCD cross communication are shown from the side (**e**) and from the top (**f**). The distance between consecutive dashed lines is $p/2$ and **m** is everywhere normal to these lines. Colors represent the selectively reflected color from CSRs as in **c**, **d**, for each reflection path. These are indicated by arrows in **e**, while the colored areas in **f** correspond to observed colored areas. Scale bar: 20 μm.

CSRs, with a character that depends sensitively on the focus, just like for the CLC precursor shell. Focusing on the equator of the shorter-pitch CSR (Fig. 5a) we see a green spiky pattern around the perimeter, while the focus at the top (b) shows mainly yellow reflections from the boundaries and centers of the FCDs.

No cross communication can be seen in (a–b) but this is a result of the short $p$ in these CSRs; the blueshift of cross communication[8] in CSRs with green retroreflection brings it into the invisible UV region. With the greater $p$ in the other CSRs the cross communication is instead deep blue, and we thus see a highly interesting new pattern arising, see Fig. 5c–d. While the previously known cross communication between CSRs is (weakly) present also here, the most striking pattern arises from cross communication within CSRs, between any FCD and its neighbors. Two examples of the resulting triangular-symmetry internal cross communication pattern are shown in panels (c–d), its origin schematically illustrated in (e–f). A full analysis of the optics of the FCD CSRs is highly challenging, not least considering the need to take the complex refraction from a continuously curving optic axis field into account[50], which is exceptionally strong here due to the highly curved FCDs. We have initiated theoretical modeling of the situation and will report on this separately.

## Application potential of FCD-templated CSRs

To make a first assessment of the application potential of the FCD-templated solid CSRs, comparing with traditional tangential-aligned CSRs as reference, we prepare macroscopic film samples by dispersing CSRs in NOA160, with refractive index $n = 1.6$ similar to the average refractive index $\bar{n}$ of the CSRs. Taking care to avoid air bubbles, we distribute about 1000 CSRs of FCD type with green retroreflection in one sample, a similar amount of CSRs of tangential type made from the same CLC mixture (containing 9% HDDA) in another, and in a third sample we randomly mix FCD and tangential CSRs made from a mixture yielding red retroreflection after polymerization. Each sample is made by spreading the CSR-in-NOA160 suspension over a square of about 1 cm by 1 cm area, and then the NOA160 is cured into solid state by UV irradiation. The resulting composites are shown, micro- and macroscopically, in Fig. 6.

Panels a–c show POM views of the same region of a sample with FCD CSRs with green retroreflection in transmission without (a) and with (b) analyzer as well as in reflection between crossed polarizers (c), whereas (d) shows a crossed-polarizer reflection POM image of a corresponding region of the sample with tangential-aligned CSRs with green retroreflection. Comparing (c) and (d), it is clear that the two CSR types yield very different patterns generated by retroreflection (cross communication is in the invisible UV range due to the short $p$ of these CSRs), the tangential CSRs exhibiting a single green spot each while the FCD CSRs show a pattern of distributed smaller green spots. Interestingly, this strong qualitative difference does not transfer to the macroscopic scale, as demonstrated in panels e–h, showing the sample with FCD CSRs on the left and the sample with tangential CSRs on the right, as a function of viewing angle under ambient light. The two samples behave qualitatively in the same way. One beneficial difference is that the colored cross section from each shell is slightly larger with FCD shells than with tangential-aligned shells, hence we may

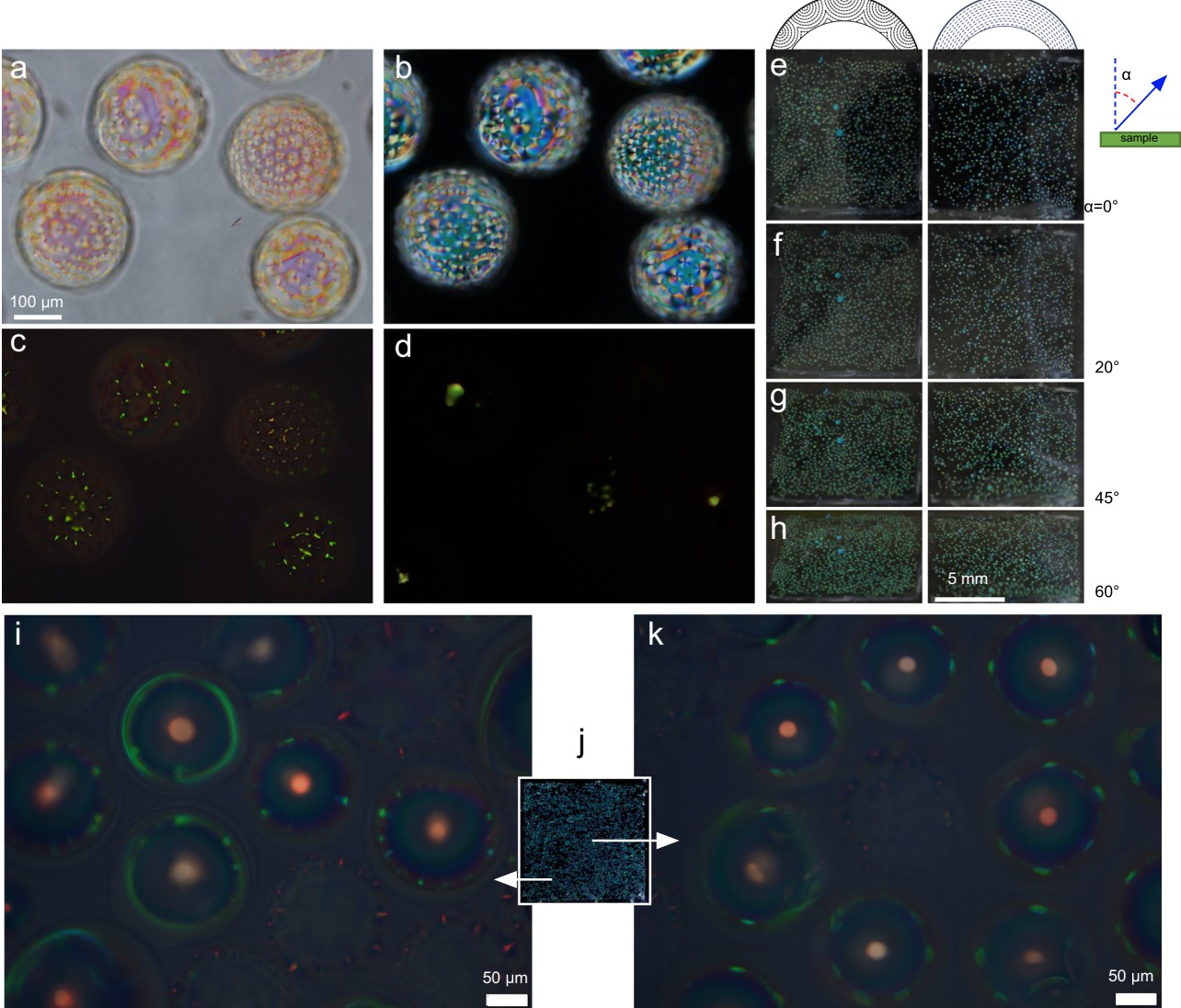

**Fig. 6 | Micro- and macroscopic reflection behavior of Cholesteric Spherical Reflectors (CSRs) polymerized at temperatures of different alignment.** Polymerized, cleaned and dried CSRs are dispersed in NOA160 binder that is photocured into a solid film. POM images are taken in transmission without analyzer (**a**) and between crossed polarizers (**b**), and in reflection between crossed polarizers (**c, d**). The CSRs in **a–c** have FCD configuration while the CSRs in (**d**) have tangential alignment. Scale bar in **a** applies also to **b–d**. In **e–h**, macroscopic photos (no polarizers, ambient illumination, black background) of the film with FCD CSRs (left column) compared with the film with tangential CSRs (right column) are shown as a function of viewing angle α with respect to the macroscopic sample normal.

Generally, α is different from the local incidence angle θ defined with respect to the helix axis **m**. The distance between consecutive dashed lines in the top schematic is *p*/2 and **m** is everywhere normal to these lines. The large cyan spots in the left sample are due to a few CSR beads dispersed together with the CSR shells. All CSRs in **a–h** have green retroreflection, while **j** shows a film with randomly mixed tangential and FCD CSRs with red retroreflection in orthogonal macroscopic view under ambient light, appearing blue-green due to cross communication and inclined illumination. Reflection POM microscopic views of two regions are shown in **i** and **k**, revealing the amplified uniqueness of the local reflection pattern. The side length of **j** is 1 cm.

expect stronger signals when these are used in markers for robotics and augmented reality[4]. Similar to the cuticle of the beetle Chrysina Gloriosa, the apparent color under diffuse light is nearly independent of viewing angle, a very useful feature that is unusual for structural color.

The complex near-field retroreflection pattern of FCD CSRs (Fig. 6c), much more information-rich than the single central retroreflection of tangential CSRs (Fig. 6d), render our new CSRs exceptionally useful in tags for anticounterfeiting and track-and-trace purposes[8,9,31,32], since even at the scale of an individual CSR, features that are unique and effectively unclonable can be detected. The ideal solution for such applications is clearly the mixing of CSR types in (i–k). Macroscopically, the film looks similar to the others, but at the microscopic scale we now see a much richer unique pattern, where

each FCD CSR disrupts most of the inter-CSR cross communication of the tangential CSRs, which thus acquire a peripheral pattern of cross communication spots with broken symmetry. This is illustrated in (i, k) using CSRs with red retroreflection, yielding visible green cross communication. Different locations of the same macroscopic sample show clearly different microscopic patterns. Interestingly, the macroscopic color under ambient light is similar to the samples with shorter-pitch CSRs, because the cross communication and inclined reflections dominate over the retroreflection[4,30]; compare (j) with (e–h).

## Shells stabilized by triblock versus random copolymers

With nematic shells of the commonly used mesogen 5CB stabilized between aqueous solutions of 87–89% hydrolyzed PVA, Durey et al.

found an analogous alignment change to what we have described above, from tangential to radial director at the shell boundaries, in the extreme vicinity of the clearing transition[36]. In that case, detection of the phenomenon required very slow heating. We noticed the same effect, with the same mesogen, over a much greater temperature range when replacing PVA with the non-ionic amphiphilic triblock copolymer Pluronic F-127[35]. Different from PVA, F-127 is designed as a surfactant, comprising a central hydrophobic polypropylene oxide (PPO) block flanked by two hydrophilic polyethylene oxide (PEO) blocks[56,57]. However, in terms of its impact on director configuration, F-127 is significantly different from the ionic surfactants that are usually used to impose normal **n** at a boundary of an LC to water. Such surfactants are expected to extend their single relatively long all-trans alkyl chain into the LC along a direction that on average is normal to the interface. F-127 and similar triblock copolymer surfactants, in contrast, cannot be expected to extend much into a liquid crystal phase, as their hydrophobic moiety is a highly flexible polymer block that would suffer a significant entropic penalty in terms of reduced conformational freedom if it were to be orientationally ordered by the LC. It is thus likely to rather adsorb onto the LC, without much penetration. The twin PEO blocks, which are expected to be fully hydrated and extend away from the LC, are also very different from the single small hydrophilic headgroup of ionic surfactants. Consequently, the impact of triblock copolymer surfactants on LC alignment is non-trivial to predict. It is important to note that the PVA that is normally used for stabilizing LC shells is actually also somewhat amphiphilic. This is because PVA is synthesized by hydrolyzing polyvinylacetate (PVAc) which is insoluble in water. Since 11–13% acetate pendants remain in the 'standard' 87–89% hydrolyzed PVA it can thus be considered amphiphilic, albeit with a random distribution of the alcohol and acetate pendants.

Given the previously observed alignment change over a broad temperature range, from tangential to normal upon heating and vice versa on cooling[35], we study the impact of F-127 on our cholesteric shells. We replace PVA by Pluronic F-127 (1 wt.%) in both inner and outer phases and we first prepare shells with the 6 wt% HDDA mixture. To our surprise, the FCD texture now appears on the surface of the shells already at room temperature. Only after placing the shells in a fridge at 5 °C they adopt tangential alignment, see Supplementary Fig. 9. The clearing point is not affected by the change of stabilizer.

We then prepare shells of the CLC base mixture stabilized by the aqueous F-127 solutions and now we find the full textural development from tangential at room temperature to FCDs as the shells are heated toward $T_{N^*I}$, as shown in Supplementary Fig. 10 and Supplementary Movie 5. The response to heating is qualitatively similar to the shells of the same CLC mixture stabilized by PVA solution, but while the PVA-stabilized shells had to be heated to $T_t \approx 71$ °C, about 1.5 °C below $T_{N^*I}$, to see the transition to FCDs (Fig. 2d–e), the F-127-stabilized shells undergo this alignment transition already at $T_t \approx 58$ °C, see Supplementary Fig. 10c–e. The overall process is slower and extends over the much larger temperature range up to the clearing transition. With a clearing temperature of $T_{N^*I} \approx 72$ °C $\approx 345$ K ($T_{N^*I}$ is not quite identical to that in the experiment with PVA as stabilizer, a consequence of slight variations in the CLC mixture composition, which is difficult to reproduce perfectly given the many components), we obtain $T_r \approx \frac{331-345}{345} \approx -0.041$. The reduced temperature range of non-tangential alignment is thus 10-fold greater with F-127 as stabilizer than when the standard PVA is used, a significant expansion. Profiting from this expanded temperature range, we use F-127-stabilized shells to study the process from the side by tilting the microscope by 90° (Supplementary Fig. 11 and Supplementary Movie 6), and we also confirm that the realignment is reversible by heating well into FCD texture but then cooling back before $T_{N^*I}$ is reached, see Supplementary Fig. 12. The tangential texture is recovered, but annealing is required to make it uniform.

**Table 1 | Overview of polymeric stabilizers compared**

| Stabilizer | Type | $P_h$ | $M$ [kg/mol] | CMT@1 wt.% [ °C] |
|---|---|---|---|---|
| PVA 1, 80% | Random | 20 | 9–10 | N.A. |
| PVA 2, 87–89% ('standard') | Random | ~12 | 13–23 | N.A. |
| PVA 3, 87–89% | Random | ~12 | 85–124 | N.A. |
| PVA 4, 98% | Random | 2 | 13–23 | N.A. |
| PVA 5, 99+% | Random | <1 | 85–124 | N.A. |
| P-123 ($EO_{20}PO_{70}EO_{20}$) | Triblock | 64 | 5.8 | 19 |
| L-64 ($EO_{13}PO_{30}EO_{13}$) | Triblock | 54 | 2.9 | 32 |
| F-127 ($EO_{106}PO_{70}EO_{106}$) | Triblock | 25 | 12.6 | 73 |
| F-68 ($EO_{76}PO_{29}EO_{76}$) | Triblock | 16 | 8.4 | 50 |
| F-108 ($EO_{136}PO_{52}EO_{136}$) | Triblock | 16 | 14.6 | 30 |

The $P_h$ and $M$ columns hold the mole-percentage of hydrophobic repeat units and the molar mass (range or average), respectively. The Critical Micelle Temperature (CMT) is the temperature below which micelles do not form for Pluronic surfactants. The data sources for PVA are the suppliers and for Pluronic surfactants they are, for P-123, L-64 and F-127:[68]; for F-68:[56]; for F-108: CMT[69] and structure[70].

Considering the dramatic difference in behavior between PVA and F-127 as stabilizer, we conduct a systematic series of reference experiments using four other PVA types as well as four other surfactants in the Pluronics series, to gain a better understanding of the impact of the chemical design as well as molar mass of polymeric stabilizers on the alignment and stability of LC shells. An overview of the different stabilizers and their characteristics is provided in Table 1. For consistency, we maintain 5 wt.% stabilizer concentration when working with PVA and 1 wt.% stabilizer concentration when using Pluronics. Starting with the PVA variation, an experiment with shells with 6 wt.% HDDA stabilized by the 80% hydrolyzed PVA 1 (the lowest hydrolysis degree that is commercially available) yields qualitatively identical results to the corresponding experiment with 'standard' PVA 2, see Supplementary Fig. 13. The temperature window of realignment appears slightly smaller but the difference is within the margin of measuring error. Shells stabilized by PVA 3, keeping the same degree of hydrolysis but increasing the molar mass by an order of magnitude, also behaves very similar to PVA 2, see Supplementary Fig. 13. The higher molar mass impacts neither shell stability nor realignment temperature range.

A dramatic difference, already on a qualitative level, is seen with PVA 4 and 5, both almost entirely hydrophilic (the 99+ quality of PVA 5 is the highest hydrolysis degree that is commercially available). Shells stabilized by PVA 5 remain tangentially aligned at all temperatures, transitioning to isotropic without any trace of FCD configuration, see Fig. 7 (snapshots from Supplementary Movie 7). With PVA 4, with 2% hydrophobic acetate groups remaining, we see some FCD formation on the thin shell side very near the clearing transition, but also here the shells remain predominantly tangential-aligned throughout the experiment, see Supplementary Fig. 13.

The last three panels of Fig. 7 show the texture of the PVA 5-stabilized shell on cooling back from isotropic. These images are interesting, first, because they reveal that the cholesteric phase forms with a different appearance, showing a highly irregular fan-shaped texture with randomly oriented in-plane helix[58]. The broad temperature range of isotropic–cholesteric phase coexistence in the multicomponent shell mixture used in this study means that the transition from isotropic to cholesteric nucleates in multiple small points in the bulk of the shell, away from the boundaries to the aqueous phases. In each nucleus the helix develops with **m** in a random direction. Once nuclei meet, a system full of grain boundaries across which **m** changes abruptly forms, which can rearrange in response to specific boundary conditions only very slowly. In Fig. 7h we see red Bragg diffraction, but the texture is still highly irregular. This emphasizes the importance of making the shells in the cholesteric state, as the shear flow and absence

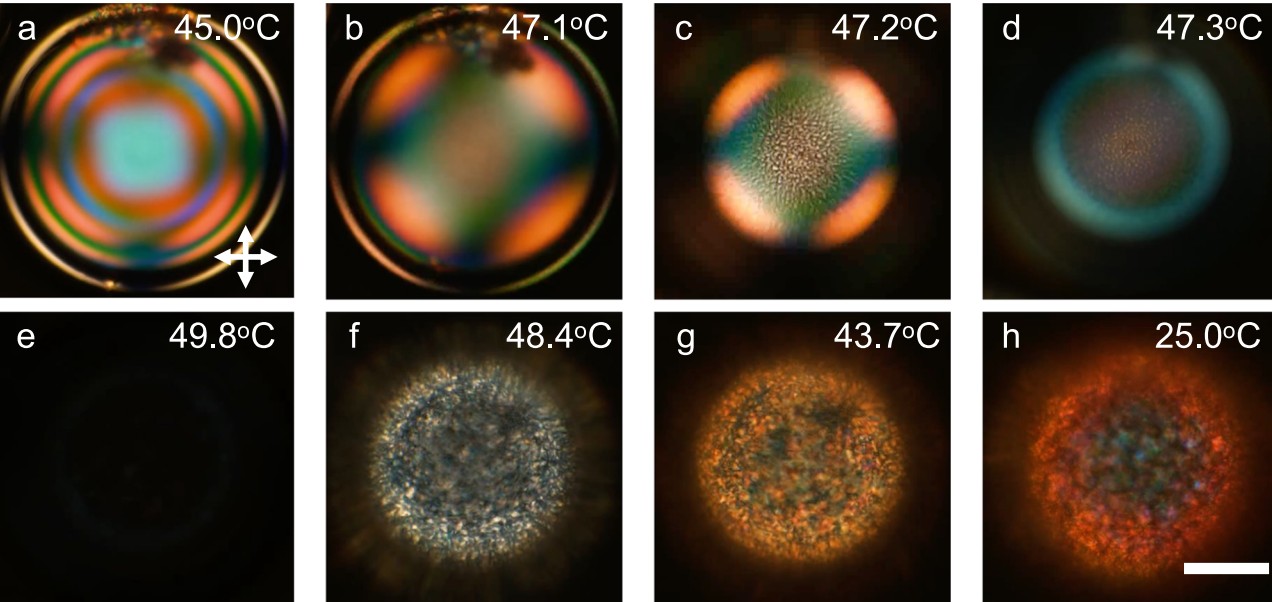

**Fig. 7 | Confirmation of lack of realignment upon heating when a cholesteric liquid crystal (CLC) shell is stabilized by fully hydrophilic stabilizer.**
**a–h** Transmission POM images (polarizer orientations indicated in **a** of a CLC shell produced with a 6 wt.% HDDA mixture and stabilized by 5 wt.% aqueous solution of PVA 5 ($M_w$ = 85–124 kg/mol, 99+ % hydrolyzed) as it is heated from room temperature to 50 °C and then cooled to 25 °C. The temperature changing rate was separated in four ranges: 5 °C/min for 23.6–43 °C, 1 °C/min for 43–45 °C, 0.3 °C/min for 45–50 °C, and 3 °C/min for 50–25 °C. The focus is at the equator of the shell in **a**, **b** and at the bottom surface in **c–h**. Scale bar: 50 μm.

of phase boundaries promote the formation of a well-aligned state as defined by the boundary conditions.

Second, the very existence of these images reveals that the shells remain stable when using 99+% hydrolyzed PVA as stabilizer. The reason that no corresponding images are included from the experiments with PVA 1–3 is that those shells always break within seconds after cooling past the isotropic–cholesteric transition. With the 98% hydrolyzed PVA 4, the shells are also stable after cooling back from isotropic, see Supplementary Fig. 13. All PVA types thus function well as a stabilizer for shells produced in the cholesteric phase, but if the phase forms in an existing shell on cooling from isotropic, only the highly hydrolyzed PVA types provide sufficient interface stabilization. Moreover, when comparing the transition with PVA 2 and 5 (Supplementary Movies 7 and 8), we note that the nucleation of LC order takes place simultaneously across the entire shell in case of 99+% hydrolyzed PVA, and the birefringence then gradually increases everywhere as the coexistence between isotropic and cholesteric phase appears to be concentric, the cholesteric gradually replacing the isotropic. When using PVA with low degree of hydrolysis, in contrast, the cholesteric phase appears in localized patches, each with high birefringence from the start. This suggests that the phase coexistence is now lateral rather than concentric, with islands of ordered phase nucleating and then growing in size, surrounded by a sea of isotropic phase. Such lateral phase separation was recently seen in case of lipids adsorbing onto nematic and smectic shells[59], driven by localized concentration of lipids as a result of the interaction with the LC phase. The amphiphilicity of PVA with less than 98% hydrolysis may give rise to a similar type of phase separation in case of the short-pitch cholesteric phase seen here, but this time with the result that the shell stability is compromised.

In the experiments comparing different Pluronic triblock copolymer stabilizers, the first striking finding is that F-108 is the only one in addition to F-127 that gives good enough shell stability to study the shell texture as a function of temperature while heating through the cholesteric phase and into the isotropic. Based on the data in Supplementary Fig. 14, we find a realignment temperature $T_t$ = 64.8 °C, a clearing transition $T_{N^*I}$ ≈ 72 °C, yielding a reduced realignment temperature $T_r$ = −0.02. The magnitude is about half that obtained for F-127, which fits well with the lower fraction of hydrophobic monomers in F-108 as indicated in Table 1. Shells prepared with F-68 or P123 in the aqueous phases generally break into droplets right after production, see Supplementary Fig. 15. When using L-64, the vast majority break but a few shells survive long enough for studying their starting texture. This turns out to be the typical fan-shaped texture of a short-pitch cholesteric with in-plane helix, i.e., it seems that the large fraction of hydrophobic block means that we here get normal alignment.

## Numerical simulation results

In what follows, we present numerically computed equilibrium profiles of CLC shells, which are results from the mathematical model defined in the Supplementary Information. The profiles are either local or global approximate minimizers of the free energy, (2) in the Supplementary Information, subject to different combinations of tangential and normal boundary conditions. We work with a fixed low temperature, $t$ = −1.79, which is simply a temperature for which the CLC system prefers an ordered (chiral) nematic state to a disordered isotropic state, on energetic grounds. This temperature has not been matched to the experiments, and we speculate that the qualitative deductions are independent of $t$, provided $t < 0$. In our simulations, we use a fixed shell size, defined by $\xi_R = \frac{1}{50}$ (see Supplementary Information for definition of $\xi_R$). This is a material-dependent and geometry-dependent length scale, but corresponds to shells which are ~50 times larger than the nematic correlation lengths. This is a small shell size, perhaps of the order of 1 micron, whereas shells used in experiments are much larger. The choice of the relatively small shell size stems from the computational expense of using a smaller value of $\xi_R$ (which would correspond to a larger shell size). However, our results do capture the tangential textures and FCDs qualitatively, including the smaller FCD size on the thinner side of asymmetric shells, as observed in the experimental snapshots in the preceding sections.

We consider symmetric and asymmetric shells separately, noting that only asymmetric shells are used in experiments. Symmetric shells

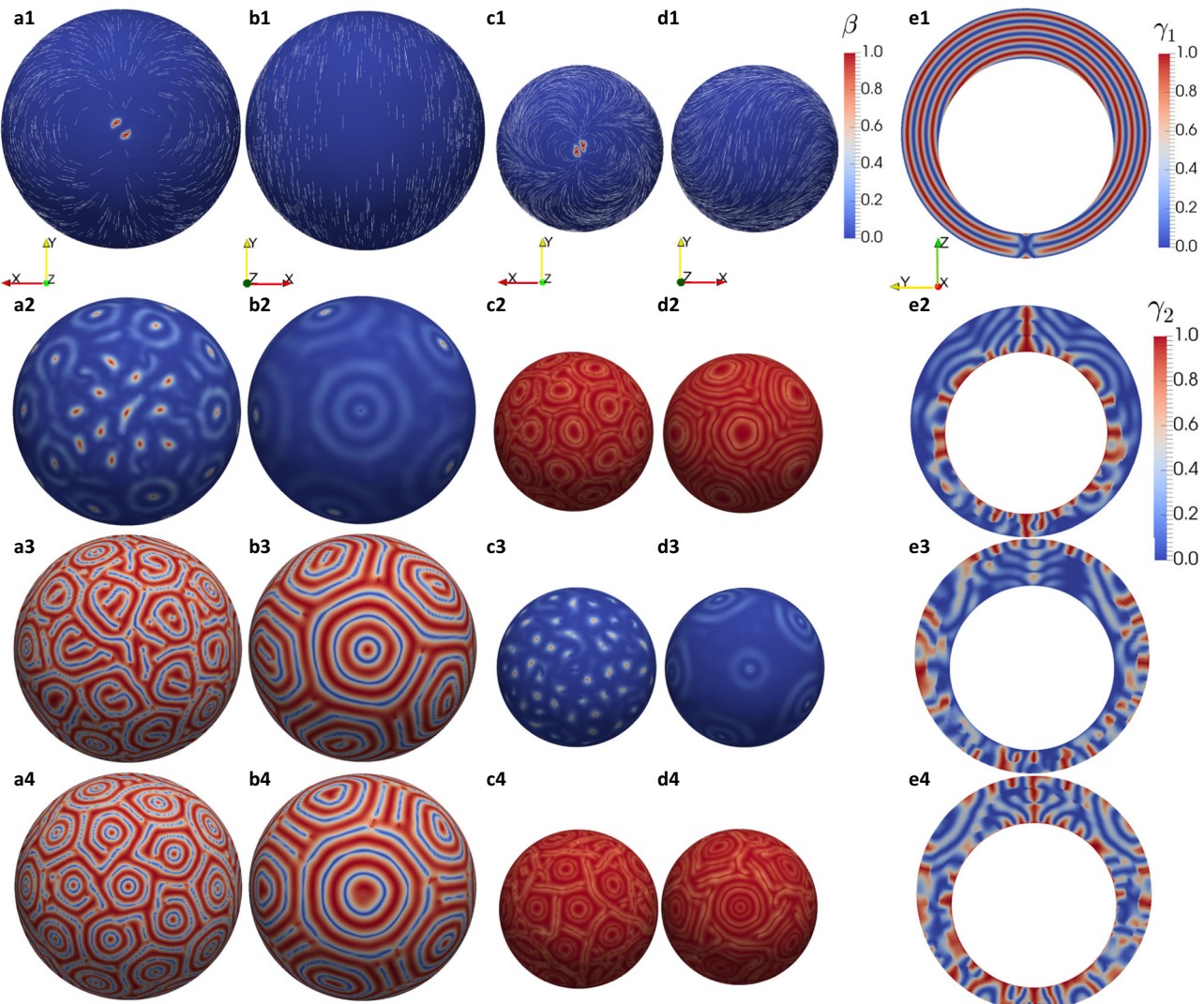

**Fig. 8 | Simulated configurations of cholesteric liquid crystal shells with different boundary conditions.** The equilibrium profiles (a1-e1) with tangential boundary conditions on the inner and outer surfaces with anchoring strength $\omega = \omega_1 = \omega_2 = 0.1$, (a2-e2) with normal on the inner surface and tangential on the outer surface $\omega = \omega_1 = \omega_2 = 0.01$, (a3-e3) with tangential on the inner surface and normal on the outer surface $\omega = \omega_1 = \omega_2 = 0.01$, (a4-e4) with normal on the inner and outer surfaces $\omega = \omega_1 = \omega_2 = 0.01$, at fixed temperature $t = -1.79$, $\xi_R = 1/50$, $\eta = 1$, $\sigma = 10\pi$, $c = 0.1$, $\rho = 0.7$. (a$i$) Bottom of outer surface; (b$i$) top of outer surface; (c$i$) bottom of inner surface; (d$i$) top of inner surface; (e$i$) cross-section; $i = 1, \cdots, 4$. The color bars label the biaxiality parameter $\beta = 1 - 6\frac{(tr\mathbf{Q}^3)^2}{(tr\mathbf{Q}^2)^3}$ in (a1-d1), $\gamma_1 = |\mathbf{n} \cdot \mathbf{e}_x|$ in (e1), and $\gamma_2 = |\mathbf{n} \cdot \mathbf{e}_\xi|$ in (a2-e4), where $\mathbf{e}_\xi$ is the unit $\xi$-direction in the bispherical coordinate defined in the Supplementary Information. The white lines in (a1–d1) label the director $\mathbf{n}$, which is the eigenvector of $\mathbf{Q}$ with the largest eigenvalue.

can be easier for visualization purposes, and some numerical results for symmetric shells are deferred to the Supplementary Information. In Fig. 8(a1-e1), we numerically compute a stable CLC configuration for an asymmetric shell, with approximately tangential boundary conditions on both the inner and outer surfaces. For the anchoring coefficients (see Supplementary Information for definition) we choose $\omega_1 = \omega_2 = 0.1$, but these are arbitrary choices which cannot be related to experimentally measurable quantities here. We speculate that this is moderately strong tangential anchoring. The biaxiality parameter is defined to be

$$\beta = 1 - 6\frac{\left(\mathrm{tr}\mathbf{Q}^3\right)^2}{|\mathbf{Q}|^6} \tag{1}$$

where $\mathbf{Q}$ is the LdG Q-tensor order parameter, such that $0 \le \beta \le 1$, and $\beta = 0$ if and only if $\mathbf{Q}$ has a pair of degenerate eigenvalues i.e. is either uniaxial or isotropic[60]. The maximal value, $\beta = 1$, occurs when $\mathbf{Q}$ has one

zero eigenvalue. We plot $\beta$ on the inner and outer surfaces, and see that $\beta = 0$ almost everywhere, except for near two point defects which are an essential topological consequence of the imposed tangential boundary conditions.

Consider Fig. 8(e1), wherein we plot a scalar quantity, $\gamma_1 = |\mathbf{n} \cdot \mathbf{e}_x|$ on a shell cross-section in the $(y, z)$-plane and $\mathbf{n}$ is the director or the eigenvector of $\mathbf{Q}$ with the largest positive eigenvalue. If $\gamma_1 = 1$, then $\mathbf{n}$ is along $\pm\mathbf{e}_x$ and if $\gamma_1 = 0$, then $\mathbf{n}$ is in the $(y, z)$-plane. We plot the contours of $\gamma_1$ to track the pitch of the CLC i.e. $\mathbf{n}$ rotates by $\pi$-radians between two red contours or between two blue contours. This plot of $\gamma_1$ corresponds to a structure that gives rise to the low-temperature images in Fig. 2a–c for which the boundary conditions are expected to be tangential on both shell surfaces. In Supplementary Fig. 16, the numerical computation is repeated on a tangential symmetric shell, and the circular twist contours of $\gamma_1$ are reproduced, as in Fig. 8(a1-e1).

In Fig. 8(a2-e2), we consider a hybrid asymmetric shell, with normal boundary conditions on the inner surface and tangential boundary conditions on the outer surface. The anchoring

coefficients are an order of magnitude smaller than the anchoring coefficients in Fig. 8(a1-e1), and we only plot a scalar quantity, defined to be $\gamma_2 = |\mathbf{n} \cdot \mathbf{e}_\xi|$, and $\mathbf{e}_\xi$ is the radial unit-vector in the unit $\xi$-direction (refer to the bispherical coordinates in the Supplementary Information). For a symmetric shell, $\mathbf{e}_\xi$ coincides with the radial unit-vector. On the inner surface, $\gamma_2$ is approximately unity, consistent with the normal boundary conditions which coerce $\mathbf{n}$ to align with the normal to the shell surface. Similarly, $\gamma_2$ almost vanishes on the outer surface, consistent with the tangential boundary conditions so that $\mathbf{n}$ is approximately in the plane of the outer shell surface. Notably, the contours of $\gamma_2$ are approximately polygonal on the inner spherical shell cross-sections, strongly reminiscent of the FCDs reported in Fig. 2, Supplementary Fig. 6 and other experimental figures of FCD shells. In fact, the FCDs in those figures most likely correspond to hybrid shells, as used in the simulation, since the change of the boundary conditions from tangential to normal upon heating occurs at different threshold temperatures according to the study of Noh et al.[35]. The numerical computation of the FCDs as stable equilibrium structures are perhaps the main theoretical result since it requires a carefully designed non-trivial initial condition for the numerical solver. We essentially need to prescribe a regular lattice of discs that tessellate the shell surfaces, and prescribe the locations of the disc centers, to imitate the FCDs in experiments.

In Fig. 8(a2-e2), the shell has normal boundary conditions on the inner surface and tangential boundary conditions on the outer surface. We see FCDs on the surface with normal boundary conditions, the inner surface. In Fig. 8(a3-e3), we switch the boundary conditions to normal on the outer surface and tangential on the inner surface, for which FCDs are then observed on the outer surface. Based on the prior study[35] this is the situation we expect to have in the current experiments. The contour plots of $\gamma_2$ demonstrate how $\mathbf{n}$ interpolates between the boundary conditions across the width of the shell. In Supplementary Fig. 17 we repeat the numerical computation on a hybrid symmetric shell, and reproduce the FCD contours of $\gamma_2$, as in Fig. 8(a3-e3). In Fig. 8(a4-e4), we plot a numerically computed stable configuration inside an asymmetric CLC shell, with normal boundary conditions on the inner and outer shell surfaces. This yields FCDs on the inner and the outer surfaces.

To summarize, our numerical results suggest that the textures in Fig. 8(a1-e1) and Supplementary Fig. 16 are found for tangentially aligned shells, with relatively strong anchoring. This is consistent with the experimental observation of tangential textures for low temperatures. The FCDs are more common with normal boundary conditions on at least one interface in Fig. 8(a2-e4), which can be associated with relatively high temperatures as further discussed below. Our model has several limitations with respect to parameter choices and the absence of a mapping between the model parameters and experimental variables, but nevertheless, the numerical simulations do capture the qualitative features of the structural transitions in CLC shells, as described in the preceding sections.

## Discussion

Our experiments clearly demonstrate that amphiphilic polymer-stabilized cholesteric LC shells in water—like non-chiral nematic shells[35,36]—will change their alignment from tangential to normal as their clearing point is approached. The realignment starts at much lower temperature for F-127- and F-108-stabilized shells than for shells stabilized by partially hydrolyzed PVA, while for fully hydrolyzed (non-amphiphilic) PVA, no realignment is observed. The key question is why this happens and which phenomena drive the alignment transition. For non-chiral shells we had previously proposed[35] that it may be related to the reduced anisotropy in interfacial tension upon decreasing orientational order parameter $S$ as the LC approaches its clearing point. We argued that the resulting weakened anchoring may render dominant the elastic energy due to the deformed director field in a tangential-

aligned shell, thus promoting a change to normal alignment. However, this argument does not hold for the cholesteric shells, where the FCD arrangement clearly has a higher elastic energy cost than the tangential configuration, and the argument is also refuted by the observation by Durey et al.[36] that the same alignment transition happens for a flat 5CB film surrounded by PVA solution, where no elastic deformation is present.

Durey et al. instead suggested that the alignment change happens because the LC layer closest to the water transitions to isotropic at a temperature slightly lower than the clearing point of the bulk LC, hence the anchoring would then be determined by an interface between nematic and isotropic LC rather than an LC–water interface, and this would promote normal alignment. The reason for the lower clearing point in the interfacial layer was suggested to be PVA partially dissolving into the LC. However, liquid crystals are poor polymer solvents since their long-range orientational order imposes an entropic penalty in terms of reduced orientational freedom. Samitsu et al. demonstrated that this entropy penalty leads to polymeric solutes in an LC to leave the ordered regimes and aggregate in disordered ones if a spatial variation of $S$ is imposed[61]. Considering this as well as the chemical incompatibility between 5CB and water soluble PVA, it seems unlikely that PVA can dissolve into nematic 5CB to an extent where it would decrease the phase transition temperatures. Moreover, the temperature range of the realignment process when using F-127 or F-108 as stabilizer appears too large for corresponding to (chiral) nematic–isotropic phase coexistence.

Considering the striking differences in behavior between amphiphilic and fully hydrophilic stabilizers, we believe that the explanation can be found by considering the impact of amphiphilicity of the stabilizing polymer. We propose to consider an aspect that was never before considered regarding LC–aqueous phase interfaces, namely the impact of the LC orientational order on the behavior of the stabilizer molecules. We assume that all stabilizers are entirely in the aqueous phases when the shell is in a (chiral) nematic state, but the hydrophobic components tend to adsorb directly onto the LC while the hydrophilic parts reach away from it to be fully hydrated. Given the amphiphilic nature of most of the polymers studied here and their consequent preference for bringing the more hydrophobic sections in contact with the LC, we should consider what impact tangential and normal boundary conditions, respectively, have on the polymer fractions in direct contact with the LC. Since the polymers are non-aromatic, the greatest chemical compatibility would arise for normal alignment, bringing the aliphatic end chains of mesogens in contact with the polymers. If $S$ is large, the resulting steric interactions between ordered mesogen end chains and hydrophobic parts of the stabilizer polymers would then reduce the conformational freedom for the latter. This would correspond to an entropy penalty that could lead to steric repulsion, just as for non-ionic surfactants that approach each other close enough for steric interaction. With the stabilizer molecule being repelled from the LC, water molecules would be present together with the stabilizer in contact with the LC, which would then instead promote tangential alignment of the LC such that the aromatic cores capable of hydrogen bonding face the water and thus lower the free energy of the interface[37]. This would thus explain the tangential alignment at low temperatures, where $S$ is high, when PVA 1–3, F-127, or F-108 are used to stabilize the LC–water interface.

Upon heating, however, $S$ is reduced, significantly so in the vicinity of the first-order transition to the isotropic state[62]. If it reaches a sufficiently low value, the entropic cost of LC-imposed ordering of the more hydrophobic parts of the stabilizer molecules may be low enough that the enthalpic gain of having aliphatic mesogen end chains in contact with the hydrophobic polymer components becomes dominant. This means that normal alignment would be favorable, and this could then explain the heating-induced alignment transition. Since PVA 2 has only a rather small fraction of 11–13% of acetate groups, and

since they additionally are randomly distributed along the overall molecule, one should expect more water to be present at the LC boundary than when F-127 is used, hence the transition only happens upon very strong reduction of $S$, explaining why we only see it rather close to the phase transition. If the LC mixture contains non-mesogenic HDDA, this may preferentially aggregate towards the interface, favoring interaction with hydrophobic stabilizer fragments, explaining the expanded temperature range of realignment for HDDA-rich shells.

With F-127 or F-108 as stabilizer, a full central block of the polymer is hydrophobic, and we can thus expect much less water in contact with the LC and strong PPO−LC contact. In this case, even a minor reduction in $S$ may be enough for the alignment change to be favorable, which would explain the much lower reduced temperature for the LC configuration to change and the greater temperature range of FCD texture, even without any HDDA in the LC mixture. At present, this is only a conjecture that needs to be corroborated in future work. We believe dedicated computer simulations and possibly neutron scattering experiments using strategically deuterated molecules may be particularly illuminating.

In a first attempt to explain also the observation that the shells break upon cooling from isotropic phase when stabilized by amphiphilic polymers, while they survive this transition without problem when stabilized by hydrophilic PVA, we speculate that the acetate pendants in incompletely hydrolyzed PVA, shunned by the water, may have been able to mix partially into the shell phase while it was in its isotropic state. This is because the absence of orientational order removes the entropic penalty arising when the solvent is ordered. As the transition to LC order takes place upon cooling, the isotropic regions which allow PVA inclusions would then be increasingly compressed by the growing LC regions that expel the polymer, leaving the ordered phase areas largely without stabilizer, thus with high interfacial tension that leads to shell breaking. Such compression of inclusions upon cooling a solvent from an isotropic to an LC state is well known, albeit in very different contexts[63-65]. The rather high concentration of PVA (5%) may amplify this effect compared to studies using the more common 1%.

Considering the impact of molar mass, we note in Table 1 that F-108 and F-127 are the only two Pluronic stabilizers in our set with a molar mass greater than 10 kg/mol, hence it appears that this may be a rough minimum for shell stability. Note that F-108 is basically identical to F-68 except that it has higher molar mass, so the difference cannot be attributed to the degree of hydrolysis. From this perspective, it is surprising that L-64, with the lowest total molar mass of all stabilizers studied, gave slightly more stable shells than with P-123 and F-68. To reconcile this, we probably need to consider not just the overall molar mass but also the molar mass of hydrophobic and hydrophilic blocks. L-64 has a close to equal fractions of hydrophilic and hydrophobic, so it may be that despite the low molar mass there is enough of each block to give some stability. With F-68, in contrast, there is only 16% hydrophobic block, amounting to about 1.3 kg/mol and this may then give too little polymer in contact with the LC. This type of reasoning could also explain the surprising observation that PVA 1 appeared to generate slightly smaller realignment temperature range than PVA 2, despite having lower degree of hydrolysis; because the molar mass of PVA 1 is only about half of that of PVA 2, the hydrophobic size per molecule of the latter is actually larger, although it is slightly stronger hydrolyzed.

In both earlier studies of the alignment transition in non-chiral nematic shells[35,36], experiments revealed that the transition takes place on the shell outside at lower temperature than at the inside, yielding an intermediate temperature range of hybrid alignment, with normal-aligned outside and tangential inside. With the cholesteric shells, the numerical simulations show that the textures to be expected on the shell outside are quite similar for hybrid and fully normal boundary conditions, but there are two experimental indications suggesting that we here see the same two-step transition, from tangential to hybrid and then from hybrid to normal, also with the cholesteric shells. First, the initial FCD texture shows strong selective reflection (Fig. 4c−d), indicating a helix that is still largely along the radial shell direction as expected for hybrid alignment (see the sketch in Fig. 1c), whereas the color almost disappears just before the shell turns isotropic (Fig. 4g), suggesting that in-plane helix orientation dominates as promoted by normal alignment on in- as well as outsides. Second, in the few cases where shells prepared with the hydrophobic-dominated Pluronics P-123 and L-64 survived long enough for us to investigate their texture, they did not show an FCD texture but instead a fan-shaped texture characteristic of short-pitch CLC phases with normal alignment at both interfaces. It is thus reasonable to assume that also for cholesteric shells, the anchoring transition takes place at lower temperature on the out- than on the inside, hence we end by discussing what could cause this difference.

In the framework of our newly proposed model, we consider that the opposite signs of curvature may be the reason. The positive curvature on the outside yields a convex interface, at which the mesogen alkyl chains in case of normal alignment have some more flexibility than at a flat interface. This would reduce the entropic penalty for a stabilizer polymer interacting with them, hence a transition to normal alignment might be expected at slightly higher bulk value of $S$ than with a flat interface, thus at lower temperature. In contrast, the negative curvature of the shell inside causes a concave interface, in which the mesogen alkyl chains in case of normal alignment are slightly squeezed together. This would reduce their flexibility and thus enhance the entropic penalty for polymers interacting sterically with them, causing repulsion even at bulk values of $S$ that are low enough for the outside to switch alignment, explaining the higher temperature at which the inside turns normal-aligned. Again, computer simulations of flexible amphiphilic polymers in an aqueous environment interacting with curved LC interfaces are needed to put this hypothesis to the test.

In conclusion, we have demonstrated that the recently discovered ability to dynamically tune the alignment of thermotropic liquid crystals in contact with aqueous phases, using an amphiphilic polymer as stabilizer, by heating them close to the clearing temperature works highly reliably for cholesteric phases exhibiting visible Bragg diffraction. This allows a controlled tuning from uniformly tangential (radial helix) to focal conic domain configuration, as demonstrated both experimentally and via numerical simulation. The effect is not specific to the way the liquid crystal is emulsified; while we here have focused on shells, with two closely spaced interfaces with the aqueous polymer solutions, we have seen during experiments that also the configuration of droplets can be tuned, their single interface with the continuous phase changing texture in a similar way. The tuning principle should thus be useful also in many other types of liquid crystal-in-water emulsion[13,16,17,25,26].

By mixing in an appropriate non-mesogenic reactive monomer to the liquid crystal, the temperatures of different alignments can be conveniently adjusted without affecting the optical properties. Moreover, since we use reactive monomers for the LC mixture, which is molded into a spherical shell form factor, we can easily make solid particles by photopolymerization, which carry over the photonic performance of the precursor LC state. When polymerizing in the focal conic state, intricate near-field selective reflection properties arise and the effective far-field reflection area per shell is increased, which may be of great value for anticounterfeiting purposes and for various photonic applications.

We propose an entirely new explanation to the origin of the temperature sensitivity of the LC configuration, which for the first time considers the impact of the nematic order on polymeric stabilizers adsorbing at the LC−water interface. When the stabilizing polymer is to some extent amphiphilic, as in the case of the polymeric surfactant

F-127 or with incompletely hydrolyzed PVA, interaction between the aliphatic chains of the LC molecules with the more hydrophobic polymer components is enthalpically favorable, but at high degree of orientational order in the LC it is entropically disfavored. For this reason, steric repulsion arises at low temperatures which leads to water being present at the LC boundary, thus leading to tangential alignment. As the temperature approaches the clearing point of the LC, however, the orientational order decreases sufficiently as to reduce the entropic penalty of close polymer–LC interaction, leading to the alignment transition. While a deeper investigation is required to test this hypothesis, the fact that LC shells stabilized by fully hydrolyzed PVA–which is fully hydrophilic–do not exhibit any alignment transition supports the model.

## Methods

The basic CLC phase used as middle phase for shell production was a mixture of (chemical structures shown in Supplementary Fig. 2) 4′-hex-5-enyloxy-biphenyl-4-carbonitrile (6OCB-1-ene[66], Synthon Chemicals), 1,4-bis-[4-(3-acryloyloxypropyloxy)benzoyloxy]-2-methylbenzene (RM257, Wilshire Technologies), S5011 (chiral dopant, HCCH, China, providing left-handed helix) and 2,2-Dimethoxy-2-phenyl-acet-ophenone (photoinitiator Irg651, Sigma Aldrich) at mass ratios shown in the top row of Supplementary Table 1. In order to reduce the clearing point and viscosity of the base mixture, varying amounts of 1,6-hexanediol diacrylate (HDDA, Sigma Aldrich) were added. All mixtures exhibit a nearly temperature-independent pitch yielding red or IR retroreflection. The mixtures were magnetically stirred in a closed vial in a water bath at 60 °C for around 5h. The thermal analysis of each mixture was carried out by differential scanning calorimetry (DSC, Mettler Toledo DSC823e, USA) as described in the Supplementary Information. The results are presented in Supplementary Fig. 3. The viscosity of each mixture was measured by a RheoSense microVISC-m, USA viscometer with a flow rate of 0.5 µL/min at 23 °C, as shown in Supplementary Fig. 4.

A nested glass capillary-based flow-focusing microfluidic device (see Supplementary Fig. 1 and a detailed account of the assembly process in the Supplementary Methods), based on a design originally introduced by Utada et al.[67], was used for the shell production. Our standard choice for isotropic inner and outer phases was a 5 wt% aqueous solution of polyvinylalcohol (PVA 2, $M_w$ = 13–23 kg/mol, 87–89% hydrolyzed, Sigma-Aldrich), flowed through the injection capillary and through the space between the cylindrical collection and the encasing square capillary as inner and outer phase, respectively. The high PVA concentration is to match the relatively high viscosity of our CLC base mixture. When other isotropic solutions were used this is noted in the main text. The CLC mixture was flowed through the space between the cylindrical injection and the square capillary. To decrease the viscosity and avoid crystallization of the middle phase, fluids and capillaries were heated if required by placing the microfluidic device on a hot stage (Linkam PE120), with the temperature set to about 10 °C below the clearing point of each mixture. An NX4-S3 (Integrated Design Tools, Inc.) high speed Movie camera mounted on a Nikon Eclipse TS100 inverted microscope was used to monitor the production process.

Shells were collected into a 20 mL vial and kept in an incubator for annealing around 12 h at the production temperature. Then, shells were either investigated optically at a POM or transferred into a petri dish and placed in a UV-curing chamber (Opsytec Dr. Gröbel Irradiation Chamber BSL-01) for about 5 min for UV polymerization, with the wavelengths 330-450 nm and light intensity of 200 mW/cm² at the sample plane. A handheld UV-LED system (30W IP66 Onforuled, China) was also used for photopolymerization while observing shells through a POM. After polymerization, solvent exchange and washing processes[24] were performed to remove all the PVA left in- and outside of shells, leaving shells dispersed in acetone for further characterization or application.

A POM (Olympus BX51, Japan) equipped with an Olympus DP73 camera (Japan) was used for optical characterization. During these studies, unpolymerized shells were kept in petri dishes or in sealed rectangular glass capillaries (0.3 mm × 3 mm cross section, CM Scientific), and a Linkam T95-PE hotstage was used to control the temperature. Macroscopic optical still images of films with embedded polymerized shells were acquired with a Canon EOS 100D digital camera. SEM imaging was carried out with a JEOL JSM-6010LA (Akishima, Japan), operated in 15–20 kV range. Samples were coated with gold (~3 nm thickness) using a Quorum Q150R ES coater (Quorum Technologies Ltd, Laughton, East Sussex, England). For SEM imaging of shell cross-sections, a rotary Microtome (Leica RM2200) was used to cut polymerized shells that had been mounted in UV-cured glue (Norland Optical Adhesive, NOA160) by applying the glue directly onto dried shells on a glass slide and shine UV light. The microtome cutting step thickness was 10 µm.

## Data availability

Source data are provided with this paper, available in a public repository at: https://osf.io/ew6ax/. Source data are provided with this paper.

## Code availability

The code used for the simulations in this paper was based on code available in public repositories at https://github.com/1Dimension/LC_Defects and at https://github.com/cen0880/CLC_Shell.

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

## Acknowledgements

This research was funded by the Luxembourg National Research Fund (FNR), grant references C20/MS/14771094 ECLIPSE (J.L.) and C21_MS_16325006 BIOFLICS (J.L.) (for the purpose of open access, the authors have applied a Creative Commons Attribution 4.0 International (CC BY 4.0) license to any Author Accepted Manuscript version arising from this submission), the Leverhulme Trust, Research Project Grant RPG-2021-401 (A.M. and Y.H.) and International Academic Fellowship IAF-2019-009 (A.M.), and the University of Strathclyde, Sir David Anderson Bequest Award (Y.H.). We thank J. Baller for the access to the DSC. We thank Yiwei Wang for sharing his numerical code for nematic shells from ref. 35.

## Author contributions

X.M. performed most experiments, some with the help of Y.G. and Y.-S.Z.; Y.-S.Z. formulated the mixtures. Y.H. carried out the numerical simulations under the guidance of A.M.; J.P.F.L. supervised the study and wrote most of the paper with input from all authors.

## Competing interests

The authors declare no competing interests.
