## [Peer Review File · Nature Communications]

REVIEWER COMMENTS

Reviewer #1 (Remarks to the Author):

This is an intriguing topic with substantial scientific content and meaningful significance. Below are a few suggestions for modification.

1. Although the content of the research is quite scientific, could the author please demonstrate some potential applications?
2. The purpose of using different types of polymers is to express various intended functions or effects.
3. Certainly, here is a summary of the mentioned articles regarding the impact of mixing different types of alignment layers on the variation of cholesteric liquid crystals (CLC), along with their potential applications:

Chen, F. L., Fan, Y. J., Lin, J. D., & Hsiao, Y. C. (2019). "Label-free, color-indicating, and sensitive biosensors of cholesteric liquid crystals on a single vertically aligned substrate." In this study, the authors explore the use of cholesteric liquid crystals (CLC) as label-free and color-indicating biosensors. The CLC is aligned on a vertically aligned substrate, and the alignment layer mixing is likely to influence the reflective properties of the liquid crystal. The article demonstrates the potential application of these CLC-based sensors for sensitive bioassays.

Hsiao, Yu-Cheng, et al. "Hybrid anchoring for a color-reflective dual-frequency cholesteric liquid crystal device switched by low voltages." *Optical Materials Express* 5.11 (2015): 2715-2720. This article discusses the use of a hybrid alignment layer in a cholesteric liquid crystal device. The authors investigate the impact of combining different anchoring techniques on the reflective properties of the liquid crystal. The resulting device can be switched using low voltages and exhibits color-reflective properties. The article suggests the application potential of such devices in display technology and other optical applications.

4. Under different conditions, phase transition temperatures can vary, and the authors should discuss this matter.

5. The size of the inner isotropic liquid and its distance from the outer shell, as well as whether these distances are consistent, can indeed impact the results. It's important to assess whether the authors have taken these factors into consideration in their study.

Reviewer #2 (Remarks to the Author):

The study authored by Ma et al. reports observations on the thermal behavior of the stabilized cholesteric liquid crystal shells. They show that when the stabilized cholesteric shells are heated from cholesteric phase to their isotropic phase, an FCD structure appears close to the isotropic transition. As the FCD structure is the result of the homeotropic anchoring of the mesogens at their interfaces with the aqueous medium and they have found also that this ordering transition is a function of the stabilizer, they attributed this ordering transition prior to the isotropic phase transition to result from the elasticity mediated interactions of the stabilizer molecules at LC interfaces. Although I think the phenomena is interesting and is of an interest of a broad audience considering its indications on the interfacial structuring of the macromolecules and their elastic interactions, I have doubts that the set of evidence presented convincingly support the major conclusions the authors made in the manuscript. Given my two major comments listed in detail below, I recommend a revision to the manuscript to clarify (that I think) the key questions before publication.

1) The authors report the formation of the FCDs in the direction of heating. While this heating occurs, the movies show that the FCDs form at a certain temperature and grow upon heating. Understanding this structuring process (nucleation, rate of formation and the sizing) may provide additional evidence on the stability and the kinetics of the of the formation of FCDs. In this context, there are two questions that I think need to be answered for further understanding.

a. Is the FCD growth a reversible process (meaning that whether the FCDs shrink when the temperature is reduced)?

b. Is the onset temperature of the ordering transition reversible? Is it heating rate independent?

I understand that the shells destabilize above T_{ni} but a simple experiment conducted below T_{ni} can simply answer these questions.

2) The authors present their evidence on the ordering transitions based on the experimental observations they made on the shells formed in PVA solutions (with and without HDDA in the LC mixture) and the shells formed in F127 solutions. With these set of experiments, they report a

stabilizer dependent onset of ordering transitions. Then, they hypothesized that the interactions of the LCs with the stabilizer macromolecules is responsible for the formation of the FCDs prior to the phase transition and made conclusions based on the three major experiments reported. Although I think their hypothesis is probably correct and the evidence are significant, given the two chemically and structurally distinct stabilizers (PVA is the partially hydrolyzed form of PVAc, and the F127 is a block co-polymer of PPO and PEO), the conclusions made may be questionable. The lack of the evidence on the formation of FCDs as a function of a systematic change in the stabilizer structure may raise doubts on the conclusions made. I would recommend the authors to present their observations made upon systematic variations either in the partial hydrolyzations of PVA or the variations of the block sizes of stabilizer PPO-PEO-PPO (somewhat derivatives of F127). In summary, a systematic change in the ordering transition temperatures upon systematic changes in the stabilizers (or at least a derivative of one of the stabilizers to compare) might provide stronger evidence on the conclusions the authors make in the manuscript.

Overall, I think the manuscript is well written and is an enjoyable read. I think the revisions based on the above comments would increase the impact of the study.

Reviewer #3 (Remarks to the Author):

The paper "Tunable templating of photonic microparticles via liquid crystal order-guided adsorption of amphiphilic polymers in emulsions" presents a thorough work on an interesting morphological transition phenomenon in cholesteric shells, in which transitions between concentric and focal-conic domain (FCD) configurations are triggered by heat-induced anchoring change respectively between the tangential and normal orientations at the shell boundaries. The work starts with the demonstration of the transition phenomenon and explanation on the structures based on optical and nano-observations, and then proceeds to optimization of conditions. The authors then presents a unique optical property, i.e. complex near-field reflections in FCD cholesteric shells. The authors also discusses based on the theoretical simulations. Although there still stand some concerns in the validity of the simulation, such as lack of size and thickness dependences, it shows qualitatively reasonable results. The text is well written and scientifically sounds. Only my concern is that in the abstract, the authors state "We utilize this phenomenon to dynamically tune the configuration of cholesteric LC shells from one with radial helix and spherically symmetric Bragg diffraction to a focal conic domain configuration with highly complex optics". But there are not very much optics. So it would nice if the authors could present some more explanations on the phenomenon, possibly based on optical modelling and simulations. In addition, the authors also proposed "The complex near-field reflections render the FCD shells exceptionally useful in tags for anticounterfeiting and track-and-trace purposes,^{6,12} the ideal solution probably being to combine FCD and tangential shells in one tag". Can the authors explain slightly more about this, possibly with more example of operation

principles? This could be helpful to catch the broad interest by readers of Nature Communications. So basically I recommend the paper for publication in Nature Communications after minor revisions to include the above.

Reviewer #4 (Remarks to the Author):

The manuscript by Lagerwall and co-workers describes the fabrication of polymerized liquid crystal (LC) microparticles, specifically cholesteric liquid crystal (CLC) shells, and how their internal structure can be tuned using temperature-dependent boundary conditions enabled by amphiphilic polymeric stabilizers. These particles exhibit either tangential or normal alignment, resulting in planar or focal conic domain (FCD) LC configurations, depending on the temperature and applied stabilizing molecules. The authors provide optical and SEM imaging for the analysis of the particles, while a microscopic setup enables the observation of their synthesis. These experimental results are complemented by theoretical modeling, which elucidates the CLC structure inside the microparticles. Overall, the presented results heavily rely on the previous studies of these authors (resulting in biased referencing with 13 self-citations). Nevertheless, I have a positive impression of the quality of the results and the novelty of stabilizer-induced CLC alignment. Below, I include some comments on the submission that could further improve the work

- During the reading, I wondered how particle size or shell thickness impacts the alignment of CLCs inside the microparticles. Do these parameters affect the behavior of CLCs under different stabilizers and temperatures? The manuscript primarily focuses on a high-chirality regime, where particle size and shell thickness are much larger than the chiral pitch. However, most photonic applications require the miniaturization of produced micro- and nanostructures. Thus, I am curious about how a low-chirality regime affects the LC alignment when the particle diameter (or shell thickness) and the chiral pitch become comparable.
- The authors focus on the influence of different stabilizers on CLC shell structures formed by a flow-focusing microfluidic device. At the same time, could they comment on how the applied stabilizing molecules may affect CLC alignment inside particles produced by suspension polymerization techniques or similar methods? For instance, references [1] and [2] demonstrate the fabrication of polymerized CLC microparticles with an onion-shaped or multidirectional asymmetric arrangement of LC layers, respectively. Perhaps the authors could expand the discussion section and consider a more generalized case of how stabilizers affect CLCs in different structures and geometries?
- I am also curious about whether the authors have experimentally observed a difference in the number of focal conical domains at the top and bottom sides of the microparticles, as shown in Fig. 6

(a3 and b3) or (a4 and b4). Apparently, according to the simulations, the number of FCDs domains is higher at the bottom side than at the top side. Have the authors observed the same effect in their samples? Perhaps microscopic imaging of the particles fixed in the UV-cured glue layer could reveal this by visualizing the sample from both sides and detecting the same particles around the recognizable cyan spots. This feature of the particles appears to be very attractive for many photonic applications, such as resonators for microlasers [1], microactuators and sensors [2], and microprobes for optical trapping and manipulation [3], [4]."

Additionally, I have some minor remarks:

- Fig. 2h lacks sufficient contrast to distinguish the boundaries of the particle against the dark background, although it's quite visible in video 1. Could the authors enhance the contrast in Fig. 2h? It would also be helpful if all images from the videos had timestamps indicating when they were captured to correlate them with the corresponding video files.
- The abbreviation 'DSC' on page 7 is not explained. It's only specified in the figure caption of Fig. S4b in the SI as 'differential scanning calorimetry.' It would be preferable to mention this abbreviation earlier in the main text when it is first introduced.
- As mentioned previously, the reference list appears biased, with many self-citations. The authors might consider revising their introduction to include coverage of other published works in the field concerning CLC microparticles and microstructures, including the works referenced below.

References:

[1] Cipparrone, G.; Mazzulla, A.; Pane, A.; Hernandez, R. J.; Bartolino, R. Chiral Self-Assembled Solid Microspheres: A Novel Multifunctional Microphotonic Device. *Advanced Materials* 2011, 23 (48), 5773–5778. <https://doi.org/10.1002/adma.201102828>.

[2] Belmonte, A.; Ussembayev, Y. Y.; Bus, T.; Nys, I.; Neyts, K.; Schenning, A. P. H. J. Dual Light and Temperature Responsive Micrometer-Sized Structural Color Actuators. *Small* 2020, 16 (1). <https://doi.org/10.1002/smll.201905219>.

[3] Donato, M. G.; Mazzulla, A.; Pagliusi, P.; Magazzù, A.; Hernandez, R. J.; Provenzano, C.; Gucciardi, P. G.; Maragò, O. M.; Cipparrone, G. Light-Induced Rotations of Chiral Birefringent Microparticles in Optical Tweezers. *Sci Rep* 2016, 6. <https://doi.org/10.1038/srep31977>.

[4] Ussembayev, Y. Y.; De Witte, N.; Liu, X.; Belmonte, A.; Bus, T.; Lubach, S.; Beunis, F.; Strubbe, F.; Schenning, A. P. H. J.; Neyts, K. Uni- and Bidirectional Rotation and Speed Control in Chiral Photonic Micromotors Powered by Light. *Small* 2023. <https://doi.org/10.1002/smll.202207095>.

1 Reviewer 1

This is an intriguing topic with substantial scientific content and meaningful significance. Below are a few suggestions for modification.

1. Although the content of the research is quite scientific, could the author please demonstrate some potential applications?

We thank the reviewer for their appraisal of our work and constructive suggestions for improvement. Indeed, the main focus at this stage has been to elucidate the process by which the cholesteric shells change their configuration in response to temperature change, but the application potential is a major motivation for our work. To better highlight this, we have in the revision added more discussion of the application possibilities, and we now use the term Cholesteric Spherical Reflector (CSR) for the polymerized solid shells to better emphasize their role in applied contexts. We now have a separate section called "Application potential of FCD-templated CSRs" and we have expanded the key application-related figure (now Fig. 6) to show both the advantages in terms of enhanced far-field reflection intensity and in terms of unique features in the near-field reflection pattern for anti-counterfeiting applications.

2. The purpose of using different types of polymers is to express various intended functions or effects.

From a practical point of view, a benefit of using different polymers for stabilization is that we can tune the temperature window of the alignment transition, making it large with a stabilizer like F127, small as with PVA that is not fully hydrolyzed, or suppress it entirely by using a purely hydrophilic stabilizer like fully hydrolyzed PVA. In this work, the main reason for studying the effect of different polymers is to better understand the origin of the realignment phenomenon.

3. Certainly, here is a summary of the mentioned articles regarding the impact of mixing different types of alignment layers on the variation of cholesteric liquid crystals (CLC), along with their potential applications:

Chen, F. L., Fan, Y. J., Lin, J. D., & Hsiao, Y. C. (2019). "Label-free, color-indicating, and sensitive biosensors of cholesteric liquid crystals on a single vertically aligned substrate." In this study, the authors explore the use of cholesteric liquid crystals (CLC) as label-free and color-indicating biosensors. The CLC is aligned on a vertically aligned substrate, and the alignment layer mixing is likely to influence the reflective properties of the liquid crystal. The article demonstrates the potential application of these CLC-based sensors for sensitive bioassays.

Hsiao, Yu-Cheng, et al. "Hybrid anchoring for a color-reflective dual-frequency cholesteric liquid crystal device switched by low voltages." *Optical Materials Express* 5.11 (2015): 2715-2720. This article discusses the use of a hybrid alignment layer in a cholesteric liquid crystal device. The authors investigate the impact of combining different anchoring techniques on the reflective properties

of the liquid crystal. The resulting device can be switched using low voltages and exhibits color-reflective properties. The article suggests the application potential of such devices in display technology and other optical applications.

We thank the reviewer for suggesting these two interesting papers. However, it is difficult to compare polymers used as alignment agents on a solid substrate to our case, where the polymer is dissolved in water and acts primarily as a stabilizer. In fact, we have difficulties to see the concrete relevance of these papers with respect to our paper. Therefore, without a better understanding of how these two papers constitute important state of the art for the work we present, we do not feel comfortable in citing them in this manuscript.

4. Under different conditions, phase transition temperatures can vary, and the authors should discuss this matter.

Yes, phase transition temperatures depend also on pressure, but usually the effect of normal atmospheric pressure variations is so small that it can safely be neglected. We do not believe that our study is an exception. In case of light-sensitive mixtures, for instance containing azo-dyes, the transition temperatures may also be light-dependent, but we do not employ such mixtures here. Other conditions that can impact the phase transition temperatures are presence of contaminants that dissolve in the liquid crystal phase. However, in our experiments the phase transition temperature (cholesteric to isotropic) remains the same for different stabilizing solutions, and varies only when we change the HDDA content. The impact of HDDA on the phase transition is already discussed.

The one experiment where we encountered a significant apparent difference is that conducted with open hot stage. As we discussed in the first version, we believed that these differences are artifacts due to incorrect temperature readings from the hot stage. We have now confirmed this assumption by measuring the sample temperature during open hot stage operation using a thermal imaging camera. Indeed, the true sample temperature measured by the camera when the control unit for the open hot stage is set to the target temperature required for observing the clearing transition is very nearly identical to the temperature at which clearing was seen when using a closed hot stage. In the revised manuscript we have added this reference experiment to the Supporting Information. We thank the reviewer for motivating us to add this experiment, as it allowed us to make a more authoritative statement about the reason for the apparent temperature discrepancy in the main paper.

5. The size of the inner isotropic liquid and its distance from the outer shell, as well as whether these distances are consistent, can indeed impact the results. It's important to assess whether the authors have taken these factors into consideration in their study.

We thank the reviewer for bringing up this important point. It is absolutely true that shell thickness, diameter and asymmetry could all be expected to influence the results. In the revised manuscript we have added a systematic and quantitative study (as far as the experimental constraints allow us) where we

have compared large and small, thick and thin shells, and we have studied the shells in more detail from the side in order to elucidate how the local thickness variations impact the focal conic domain characteristics. The results of these new studies have been integrated throughout the revision, which we believe gives a much more complete picture. For instance, we find that the alignment transition happens at slightly lower temperature on heating in a shell with low average thickness compared to thicker shells, and the focal conic domain diameter decreases from the thick to the thin side within a single shell.

2 Reviewer 2

The study authored by Ma et al. reports observations on the thermal behavior of the stabilized cholesteric liquid crystal shells. They show that when the stabilized cholesteric shells are heated from cholesteric phase to their isotropic phase, an FCD structure appears close to the isotropic transition. As the FCD structure is the result of the homeotropic anchoring of the mesogens at their interfaces with the aqueous medium and they have found also that this ordering transition is a function of the stabilizer, they attributed this ordering transition prior to the isotropic phase transition to result from the elasticity mediated interactions of the stabilizer molecules at LC interfaces. Although I think the phenomena is interesting and is of an interest of a broad audience considering its indications on the interfacial structuring of the macromolecules and their elastic interactions, I have doubts that the set of evidence presented convincingly support the major conclusions the authors made in the manuscript. Given my two major comments listed in detail below, I recommend a revision to the manuscript to clarify (that I think) the key questions before publication.

1) The authors report the formation of the FCDs in the direction of heating. While this heating occurs, the movies show that the FCDs form at a certain temperature and grow upon heating. Understanding this structuring process (nucleation, rate of formation and the sizing) may provide additional evidence on the stability and the kinetics of the of the formation of FCDs. In this context, there are two questions that I think need to be answered for further understanding.

a. Is the FCD growth a reversible process (meaning that whether the FCDs shrink when the temperature is reduced)?

We thank the reviewer for their pertinent remarks and questions, which we agree are very interesting. Before commenting in detail, allow us to stress the importance of the fact that the appearance of FCDs here is triggered by a change of boundary conditions, which is also rather gradual. This means that the formation process is not a phase transition and it is not obvious that it can be described by a distinct nucleation and growth process. Rather, it appears to involve a complex restructuring where it is difficult to identify an FCD until the process is almost complete, which can be seen quite nicely in the new Supporting Video 6 where we show the alignment transition taking place in shells filmed from the side, thus showing the behavior along the full thickness gradient.

It is difficult to judge if there even is a well-defined growth process that can be quantitatively characterized, or if it should rather be characterized in terms of annealing. What we can do in addition to filming it from the side is to confirm that, indeed, the process is reversible. We have now conducted multiple experiments where we heated F127-stabilized shells (this gives the maximum transition temperature range) at different rates into the FCD formation range without going to isotropic, stopped and waited for a few minutes, and then cooled down. This always brought the shell back to a planar-aligned state, clearly proving that the process is reversible. However, the planar-aligned state is not very well aligned directly after the cooling process, again suggesting that the realignment process is not a smooth one that can be followed quantitatively, but it is a rather dramatic case of restructuring, with annealing required after the alignment transition, in particular on cooling from the FCD state to a fully tangential configuration, to achieve good alignment.

b. Is the onset temperature of the ordering transition reversible? Is it heating rate independent?

The experiment just mentioned (Fig. S11 in the revised Supplementary Information) showed that faster heating appears to give slightly higher transition temperatures, but this is likely mainly a result of the inertia in changing the actual temperature within the shell; at high heating rates there is a lag between the hot stage temperature and the shell temperature. Our conclusion is that the onset temperature is well defined with respect to temperature in the LC, any variations being due primarily to experimental artifacts.

I understand that the shells destabilize above T_{ni} but a simple experiment conducted below T_{ni} can simply answer these questions.

Indeed, this is what we did, see Fig. S11.

2) The authors present their evidence on the ordering transitions based on the experimental observations they made on the shells formed in PVA solutions (with and without HDDA in the LC mixture) and the shells formed in F127 solutions. With these set of experiments, they report a stabilizer dependent onset of ordering transitions. Then, they hypothesized that the interactions of the LCs with the stabilizer macromolecules is responsible for the formation of the FCDs prior to the phase transition and made conclusions based on the three major experiments reported. Although I think their hypothesis is probably correct and the evidence are significant, given the two chemically and structurally distinct stabilizers (PVA is the partially hydrolyzed form of PVAc, and the F127 is a block co-polymer of PPO and PEO), the conclusions made may be questionable. The lack of the evidence on the formation of FCDs as a function of a systematic change in the stabilizer structure may raise doubts on the conclusions made. I would recommend the authors to present their observations made upon systematic variations either in the partial hydrolyzations of PVA or the variations of the block sizes of stabilizer PPO-PEO-PPO (somewhat derivatives of F127). In summary, a systematic change in the ordering transition tempera-

tures upon systematic changes in the stabilizers (or at least a derivative of one of the stabilizers to compare) might provide stronger evidence on the conclusions the authors make in the manuscript.

It is indeed highly desirable to acquire more data on the impact of the polymer structure, but it is not entirely easy to achieve. The reviewer's idea to systematically vary the choice of Pluronic stabilizer is excellent and we have thus purchased a strategic set for new experiments to bring further evidence. With PVA there are less options for variations, but we could at least obtain 80% hydrolyzed PVA, thus slightly lower degree of hydrolysis than in the standard PVA, and we have compared low and high molar mass for the 85-87% and 98-99% hydrolyzed PVA. For the revision we have added a table where we compare all the different stabilizers (10 in total) and we discuss the data obtained with each one, showing the data in the Supporting Information. The new data generally support our original conjecture. A complication is that the shell stability varies between the stabilizers, particularly poor results arising from the combination of low molar mass and high fraction of hydrophobic monomers.

Overall, I think the manuscript is well written and is an enjoyable read. I think the revisions based on the above comments would increase the impact of the study.

We thank the reviewer for this encouraging conclusion and for the very helpful advice for further improving our paper. We hope that the new results in the revision will be considered satisfactory.

3 Reviewer 3

The paper "Tunable templating of photonic microparticles via liquid crystal order-guided adsorption of amphiphilic polymers in emulsions" presents a thorough work on an interesting morphological transition phenomenon in cholesteric shells, in which transitions between concentric and focal-conic domain (FCD) configurations are triggered by heat-induced anchoring change respectively between the tangential and normal orientations at the shell boundaries. The work starts with the demonstration of the transition phenomenon and explanation on the structures based on optical and nano-observations, and then proceeds to optimization of conditions. The authors then presents a unique optical property, i.e. complex near-field reflections in FCD cholesteric shells. The authors also discusses based on the theoretical simulations. Although there still stand some concerns in the validity of the simulation, such as lack of size and thickness dependences, it shows qualitatively reasonable results. The text is well written and scientifically sounds.

Only my concern is that in the abstract, the authors state "We utilize this phenomenon to dynamically tune the configuration of cholesteric LC shells from one with radial helix and spherically symmetric Bragg diffraction to a focal conic domain configuration with highly complex optics". But there are not very much optics. So it would nice if the authors could present some more explanations on the phenomenon, possibly based on optical modelling and simulations.

We thank the reviewer for the positive assessment of our manuscript and the helpful suggestions for further improvement. Within a collaboration with Prof. Gerd Schröder-Turk, Murdoch University, such a simulation is being planned. However, the system is not trivial to simulate, hence it is unfortunately impossible to incorporate this into the revision. Instead we have conducted more careful optical characterization of the polymerized focal conic domain CSRs, also adding experiments on CSRs with red retroreflection after polymerization. The new data reveal some quite exotic optical features related to photonic cross communication between different focal conic domains within the same CSR, as well as between adjacent CSRs, and we have added a first analysis of these features and their origin around the new Fig. 5. We hope that this adds sufficient details on the novel optics at this stage. The simulations with Prof. Turk are scheduled for March–May 2024 so we hope to be able to report on their outcome in the summer–fall.

In addition, the authors also proposed "The complex near-field reflections render the FCD shells exceptionally useful in tags for anticounterfeiting and track-and-trace purposes,^{6,12} the ideal solution probably being to combine FCD and tangential shells in one tag". Can the authors explain slightly more about this, possibly with more example of operation principles? This could be helpful to catch the broad interest by readers of Nature Communications. So basically I recommend the paper for publication in Nature Communications after minor revisions to include the above.

We are glad to read that the reviewer recommends our paper for publication pending a satisfactory revision, and we agree that a deeper discussion of the use in authentication for anticounterfeiting and track-and-trace application is warranted. When we wrote the first version of the paper, we were very much focused on explaining the fundamental phenomenon, and therefore we did not pay as much attention to elaborate on the application possibilities. As mentioned above, we have now added a subsection devoted to these opportunities, expanding the relevant figure (now Fig. 6) with new experimental data, also in response to the comments of Reviewer 1, to include a microscopic investigation of an authentication 'tag' that comprises both regular and FCD CSRs. This clearly shows a much more intricate optical signature with enhanced unique features, since also the above-mentioned cross communication between FCDs in one and the same CSR corresponds to a unique and effectively unclonable fingerprint, given the variability of FCD size as well as of the way in which the CSRs puncture during washing after polymerization, rendering it practically impossible to reproduce a particular CSR type. With the FCD CSRs, in principle already a single CSR can thus provide an unclonable fingerprint, allowing the ID tag to be made exceptionally small. By combining FCD and fully tangential CSRs in a tag with multiple CSRs, a beneficial combination of unique features on scales ranging from μm to mm arises. We hope that this figure and the new text discussing these features will provide a clear idea of why and how the FCD CSRs allow a further boost in security of authentication tags based on CSRs.

4 Reviewer 4

The manuscript by Lagerwall and co-workers describes the fabrication of polymerized liquid crystal (LC) microparticles, specifically cholesteric liquid crystal (CLC) shells, and how their internal structure can be tuned using temperature-dependent boundary conditions enabled by amphiphilic polymeric stabilizers. These particles exhibit either tangential or normal alignment, resulting in planar or focal conic domain (FCD) LC configurations, depending on the temperature and applied stabilizing molecules. The authors provide optical and SEM imaging for the analysis of the particles, while a microscopic setup enables the observation of their synthesis. These experimental results are complemented by theoretical modeling, which elucidates the CLC structure inside the microparticles. Overall, the presented results heavily rely on the previous studies of these authors (resulting in biased referencing with 13 self-citations). Nevertheless, I have a positive impression of the quality of the results and the novelty of stabilizer-induced CLC alignment. Below, I include some comments on the submission that could further improve the work

We thank the reviewer for the overall positive comments and apologize for the biased referencing. Our introduction and discussion focused strongly on cholesteric *shells*, where our group has been strongly active for about a decade, which means that we do base many of our methods on prior work from our group, explaining some of the self citations. However, there are many excellent works on cholesteric droplets that we did not cite in the first version, and we have now added a selection of relevant papers in that body of literature to our bibliography. We have also added a citation to a very nice simulation work dealing with liquid crystal shells that we forgot to cite in the first version, and we removed self citations if they are not essential for putting forth our arguments or explaining our methodology.

During the reading, I wondered how particle size or shell thickness impacts the alignment of CLCs inside the microparticles. Do these parameters affect the behavior of CLCs under different stabilizers and temperatures?

As mentioned above, we have added an investigation of these aspects in the revision, which is as systematic and quantitative as the experimental constraints allow.

The manuscript primarily focuses on a high-chirality regime, where particle size and shell thickness are much larger than the chiral pitch. However, most photonic applications require the miniaturization of produced micro- and nanostructures. Thus, I am curious about how a low-chirality regime affects the LC alignment when the particle diameter (or shell thickness) and the chiral pitch become comparable.

In practice, it is difficult to make CSRs much smaller than 50–100 μm in diameter, hence the applications we have in mind are primarily those where such a particle size is acceptable, e.g., in authentication solutions (see above). While we believe their photonic properties can be applied also in other ways at this scale,

applications where the particle size approaches the single-wavelength regime are not realistic. Therefore, the practical low-chirality regime of CSRs rather corresponds to situations where the helix pitch is extended to several microns. We have investigated such shells and, to our surprise, we found that they often end up less stable than the shells with sub-micron pitch. The reason for this is still not confirmed, but we are in the process of testing a number of hypotheses in collaboration with Prof. Ralf Stannarius and Prof. Kirsten Harth, who are experts in the rupture processes of bubbles and shells of liquid crystals. The results of these studies will be reported separately.

Since our shells are asymmetric, the thinnest region of a shell is often comparable to the pitch even with the short-pitch cholesterics used in our current study. To investigate the behavior near the thinnest point we have conducted experiments with the microscope tilted by 90° , such that we can observe the liquid crystal state shells from the side, allowing a study of the behavior as a function of local shell thickness. Additionally, we added optical microscopy images of polymerized CSRs viewed from the side. We then note that the FCDs get continuously smaller from the thick to the thin side, eventually disappearing entirely. The optical resolution does not allow us to accurately measure the exact thickness when this happens, unfortunately, but qualitatively the behavior is clear. We have added these experiments to the revised manuscript as well as a corresponding discussion.

The authors focus on the influence of different stabilizers on CLC shell structures formed by a flow-focusing microfluidic device. At the same time, could they comment on how the applied stabilizing molecules may affect CLC alignment inside particles produced by suspension polymerization techniques or similar methods? For instance, references [1] and [2] demonstrate the fabrication of polymerized CLC microparticles with an onion-shaped or multidirectional asymmetric arrangement of LC layers, respectively. Perhaps the authors could expand the discussion section and consider a more generalized case of how stabilizers affect CLCs in different structures and geometries?

Indeed, the dependence of alignment on the type of stabilizer and the order parameter of the LC should not be limited to a certain method for emulsifying the liquid crystal, but should apply generally. A significant difference between shells (as studied in our paper) and droplets (studied in the two mentioned papers) is that the latter have only one interface where the stabilizers come into play while the former have two such interfaces, typically rather closely spaced. We have not systematically studied droplets, but shells may break during experiments, collapsing into droplets which are then observed simultaneously with the shells. Our experience is that the realignment upon temperature variations is seen also in droplets. We have now added a comment about this and we refer to the two suggested papers (and others) where droplets are made, pointing out that also such droplets could have the alignment tunable by temperature when utilizing this phenomenon.

I am also curious about whether the authors have experimentally observed a

difference in the number of focal conical domains at the top and bottom sides of the microparticles, as shown in Fig. 6 (a3 and b3) or (a4 and b4). Apparently, according to the simulations, the number of FCDs domains is higher at the bottom side than at the top side. Have the authors observed the same effect in their samples? Perhaps microscopic imaging of the particles fixed in the UV-cured glue layer could reveal this by visualizing the sample from both sides and detecting the same particles around the recognizable cyan spots. This feature of the particles appears to be very attractive for many photonic applications, such as resonators for microlasers [1], microactuators and sensors [2], and microprobes for optical trapping and manipulation [3], [4].”

As mentioned above, we have now conducted experiments where shells are studied from the side, allowing us to correlate the FCD size with the shell thickness. Indeed, the experiments qualitatively replicate the simulations in the sense that FCDs get smaller with decreasing shell thickness. A quantitative comparison is difficult since the simulated shells are two orders of magnitude smaller than the real ones, in order to make the simulations tractable. We thank the reviewer for the suggestions of applications of this aspect, and we have added these points together with citations of the papers in question.

Additionally, I have some minor remarks:

Fig. 2h lacks sufficient contrast to distinguish the boundaries of the particle against the dark background, although it’s quite visible in video 1. Could the authors enhance the contrast in Fig. 2h? It would also be helpful if all images from the videos had timestamps indicating when they were captured to correlate them with the corresponding video files.

We have digitally increased the brightness of the photo in 2h to allow detection of the shell boundaries, pointing out this manipulation and the reason for it in the caption. Concerning the time stamps of the still frames from videos, we fear they may have a cluttering effect if added directly into each figure, and we therefore added this information in the video captions in the Supporting Information.

The abbreviation ‘DSC’ on page 7 is not explained. It’s only specified in the figure caption of Fig. S4b in the SI as ‘differential scanning calorimetry.’ It would be preferable to mention this abbreviation earlier in the main text when it is first introduced.

We thank the reviewer for noting this omission and pointing it out to us. We have now given the full name the first time the abbreviation is used, in the main text of the paper.

As mentioned previously, the reference list appears biased, with many self-citations. The authors might consider revising their introduction to include coverage of other published works in the field concerning CLC microparticles and microstructures, including the works referenced below.

References: [1] Cipparrone, G.; Mazzulla, A.; Pane, A.; Hernandez, R. J.; Bartolino, R. Chiral Self-Assembled Solid Microspheres: A Novel Multifunctional Microphotonic Device. *Advanced Materials* 2011, 23 (48), 5773–5778.

<https://doi.org/10.1002/adma.201102828>. [2] Belmonte, A.; Ussembayev, Y. Y.; Bus, T.; Nys, I.; Neyts, K.; Schenning, A. P. H. J. Dual Light and Temperature Responsive Micrometer-Sized Structural Color Actuators. *Small* 2020, 16 (1). <https://doi.org/10.1002/sml.201905219>. [3] Donato, M. G.; Mazzulla, A.; Pagliusi, P.; Magazzù, A.; Hernandez, R. J.; Provenzano, C.; Gucciardi, P. G.; Maragò, O. M.; Cipparrone, G. Light-Induced Rotations of Chiral Birefringent Microparticles in Optical Tweezers. *Sci Rep* 2016, 6. <https://doi.org/10.1038/srep31977>. [4] Ussembayev, Y. Y.; De Witte, N.; Liu, X.; Belmonte, A.; Bus, T.; Lubach, S.; Beunis, F.; Strubbe, F.; Schenning, A. P. H. J.; Neyts, K. Uni- and Bidirectional Rotation and Speed Control in Chiral Photonic Micromotors Powered by Light. *Small* 2023. <https://doi.org/10.1002/sml.202207095>.

We fully agree with this assessment and apologize for not being more careful when selecting papers for the bibliography. We believe the revised bibliography is more balanced, and it includes all these suggested papers.

REVIEWERS' COMMENTS

Reviewer #2 (Remarks to the Author):

The authors have addressed the comments adequately and I recommend the manuscript for publication in Nature Communications.

Reviewer #3 (Remarks to the Author):

I think the authors well answered to the concerns by the reviewers. So the paper can be acceptable.

Reviewer #4 (Remarks to the Author):

I have thoroughly reviewed the revised manuscript and carefully considered the authors' responses to the reviewers' comments. I am pleased to note that the authors have addressed all of my suggestions, resulting in a notable improvement compared to the previous version. I find the manuscript to be in good shape with no additional comments from my end, and I recommend its publication in Nature Communications.